# Prior expectations guide multisensory integration during face-to-face communication

**Giulia Mazzi**[1,☯*], **Ambra Ferrari**[1,2,3☯], **Maria Laura Mencaroni**[1,4], **Chiara Valzolgher**[1], **Mirko Tommasini**[1,4], **Francesco Pavani**[1,5], **Stefania Benetti**[1,5]

1 Center for Mind/Brain Sciences (CIMeC), University of Trento, Rovereto, Italy, 2 Max Plank Institute for Psycholinguistics, Nijmegen, The Netherlands, 3 Donders Institute for Brain, Cognition and Behaviour, Radboud University, Nijmegen, The Netherlands, 4 Department of Psychology and Cognitive Sciences, University of Trento, Rovereto, Italy, 5 Centro Interuniversitario di Ricerca "Cognizione, Linguaggio e Sordità", CIRCLeS., Trento, Italy

☯ These authors contributed equally to this work.
* giulia.mazzi@unitn.it

## Abstract

Face-to-face communication relies on the seamless integration of multisensory signals, including voice, gaze, and head movements, to convey meaning effectively. This poses a fundamental computational challenge: optimally binding signals sharing the same communicative intention (e.g., looking at the addressee while speaking) and segregating unrelated signals (e.g., looking away while coughing), all within the rapid turn-taking dynamics of conversation. Critically, the computational mechanisms underlying this extraordinary feat remain largely unknown. Here, we cast face-to-face communication as a Bayesian Causal Inference problem to formally test whether prior expectations arbitrate between the integration and segregation of vocal and bodily signals. Specifically, we asked whether there is a stronger prior tendency to integrate audiovisual signals that convey the same communicative intention, thus establishing a crossmodal pragmatic correspondence. Additionally, we evaluated whether observers solve causal inference by adopting optimal Bayesian decision strategies or non-optimal approximate heuristics. In a spatial localization task, participants watched audiovisual clips of a speaker where the audio (voice) and the video (bodily cues) were sampled either from congruent positions or at increasing spatial disparities. Crucially, we manipulated the pragmatic correspondence of the signals: in a communicative condition, the speaker addressed the participant with their head, gaze and speech; in a non-communicative condition, the speaker kept the head down and produced a meaningless vocalization. We measured audiovisual integration through the ventriloquist effect, which quantifies how much the perceived audio position is misplaced towards the video position. Combining psychophysics with computational modelling, we show that observers solved audiovisual causal inference using non-optimal heuristics that nevertheless approximate optimal Bayesian inference with high accuracy. Remarkably, participants showed a stronger tendency to integrate

**Data availability statement:** All materials, data and code relevant to replicating the current findings are available on the Open Science Framework (https://osf.io/596wt/).

**Funding:** This work was supported by the University of Trento DM737/21-StartGrant-R06 awarded to SB and by a grant from the Velux Foundation (project 1439) awarded to FP. SB and FP were also supported by two "Progetto di Rilevante Interesse Nazionale (PRIN) from the Italian Ministry of Education, University and Research (MUR-PRIN 2017 n20177894ZH and Next Generation EU MUR-PRIN 2022 n.2022FT8HNC). FP and CV were supported by a grant from the Velux Foundation (Project 1439). The funders had no role in study design, data collection and analysis, decision to publish, or preparation of the manuscript.

**Competing interests:** The authors have declared that no competing interests exist.

vocal and bodily information when signals conveyed congruent communicative intent, suggesting that pragmatic correspondences enhance multisensory integration. Collectively, our findings provide novel and compelling evidence that face-to-face communication is shaped by deeply ingrained expectations about how multisensory signals should be structured and interpreted.

## Author summary

Face-to-face communication is complex: what we say is coupled with bodily signals, offset in time, which may or may not work in concert to convey meaning. Yet, the brain rapidly determines which multisensory signals belong together and which, instead, must be kept apart, suggesting that prior expectations play a crucial role in this decision-making process. Here, we directly tested this hypothesis using Bayesian computational modelling, which allows for isolating the contribution of prior expectations and sensory uncertainty on the final perceptual decision. We found that people have a stronger prior tendency to combine vocal and bodily signals when they convey the same communicative intent (i.e., the speaker addresses the observer concurrently with their head, gaze and speech) relative to when this correspondence is absent. Thus, the brain uses prior expectations to bind multisensory signals that carry converging communicative meaning. These findings provide key insight into the sophisticated mechanisms underpinning efficient multimodal communication.

## Introduction

Language evolved, is learned, and is often used in face-to-face interactions, where speech is coupled with multiple concurrent layers of bodily information, such as head movements and gaze shifts [1–5]. Some of these multisensory signals are directly relevant to the communicative exchange. For example, imagine you enter a busy restaurant and a waiter turns his body and head toward you, addresses you with a gaze shift and offers his help to find a seat. All these inputs, combined, signal the intention to communicate and thus carry congruent pragmatic meaning [1,4,6–9]. As such, it seems reasonable to assume that these signals belong to the same social act and should be integrated. Conversely, some multimodal signals may be tangential to the communicative exchange: they simply co-occur with each other but are not linked by the same underlying intention. For example, imagine that the waiter breaks into a brief cough: this signal is concurrent yet unrelated to the message being delivered and should therefore be segregated. Thus, temporal synchrony alone is not a reliable cue for integration because unrelated signals can be synchronous, while related signals can be temporally misaligned [4,10]. To further complicate the decision-making process, each conversational partner is taxed by fast turn-taking dynamics [11]. Yet, despite these critical constraints, we process multimodal communicative signals

faster and more accurately than speech alone [12–17]. How do we achieve such an extraordinary feat so seamlessly and efficiently? Current behavioural evidence cannot address this question because it is descriptive: it indirectly suggests that multimodal signals facilitate human communication. Despite decades of research on multimodal integration and speech perception, the computational mechanisms underpinning seamless face-to-face communicative interactions remain largely unexplored.

Research on the normative principles of multisensory integration offers a powerful, computationally explicit platform for directly addressing this crucial, outstanding point. Importantly, we must first ask: what is the core computational challenge that we face when parsing the conversational scene? As exemplified above, face-to-face communication implies the fundamental binding problem of multisensory perception [5]: individuals must optimally integrate signals that come from a common source and segregate those from independent sources [18]. Critically, observers cannot directly perceive the underlying causal structure (i.e., the underlying causal relationship between signals). They must infer it based on the available crossmodal correspondences. Beyond the realm of face-to-face interactions, observers typically exploit a vast set of correspondences that includes temporal synchrony [19–26], spatial congruence [27–29], semantic correspondences [30–34] and synesthetic correspondences [35–37]. The (in)congruence of such cues modulates the strength of the observer's common-cause assumption, that is the prior belief that signals should be integrated into one unified percept [37–42]. More generally, the common cause prior is supported by observers' expectations [37,39], which in turn may depend on prior experience [43–47], training [48,49], ontogenetic [50] and phylogenetic [51] factors. Hierarchical Bayesian Causal Inference [18,52–56] provides a powerful normative strategy to arbitrate between the integration and segregation of multisensory inputs. For example, in a spatial localization task (e.g., judging the speaker's position), observers should integrate auditory (e.g., voice position) and visual (e.g., body position) information according to the underlying causal structure, guided by the amount of audiovisual spatial disparity and the strength of the common cause prior [57–59]. Under a common source, observers should perform multisensory integration following the Forced Fusion principle, i.e., combining the unisensory components weighted by their relative sensory reliabilities [60–62]. Under independent sources, the unisensory signals should be processed independently. To compute a final response that accounts for causal uncertainty, observers should apply a final decision function, such as model averaging [52,53]: the estimates (e.g., spatial location) under the two potential causal structures are combined weighted by the posterior probabilities of these structures.

In face-to-face interactions, a crucial causal structure to infer is the speaker's intention: for instance, whether the intention is to communicate (e.g., looking at the addressee while speaking) or otherwise (e.g., looking away while coughing). Audiovisual signals that are shared by the same communicative intention carry a potent crossmodal pragmatic correspondence [1,4,6–9]. Such correspondence may bolster the common cause prior and thereby guide the fast and efficient segmentation of the conversational scene into coherent, meaningful units. However, a simpler computational mechanism is also conceivable. Under the Forced Fusion account [60–62], observers mandatorily perform reliability-weighted integration without considering causal uncertainty. Here, information may be filtered according to its salience and relevance [63]: communicative bodily movements (e.g., facing and gazing at someone) are more perceptually salient [64–70] and more self-relevant [71,72] than non-communicative ones (e.g., looking away); thus, they may attract more attention and thereby guide the parsing of the conversational scene. Consequently, several key questions remain unanswered. First, which computational principles govern multisensory integration during face-to-face communication? Second, is integration stronger for communicative than non-communicative signals? If so, why? On the one hand, head and gaze movements towards the observer may attract more attention than looking away, decrease the noise (i.e., increase the reliability) of the attended visual information, and thereby increase its weight in the fusion process under both Forced Fusion and Bayesian Causal Inference accounts [73]. On the other hand, crossmodal pragmatic correspondences among communicative signals may reinforce the prior belief that these signals come from a common cause, as uniquely captured by Bayesian Causal Inference. Critically, observers may also adopt non-optimal heuristic approximations to causal inference [55].

In two consecutive experiments, we formally tested these competing hypotheses by combining psychophysics and computational modelling of human behaviour. We presented participants with communicative and non-communicative audiovisual stimuli and quantified audiovisual integration through the spatial ventriloquist effect [74]. Since non-communicative crossmodal correspondences (e.g., spatial congruency) are known to drive crossmodal binding and influence perceptual inference [27–29], non-communicative stimuli provided a meaningful baseline within a multisensory integration framework. The core question we aimed to address is whether the addition of communicative intent strengthens the tendency toward integration, compared to when the signals are non-communicative but closely matched in timing and spatial configuration. Our findings demonstrate that multisensory integration was stronger for communicative than non-communicative signals and suggest that pragmatic correspondences strengthened the a priori assumption that multisensory signals belong together. Interestingly, observers' behaviour was best captured by non-optimal decision strategies that nevertheless approximate optimal Bayesian inference with high accuracy. Collectively, the present findings offer novel and direct insight into the computational mechanisms driving the efficiency of multimodal communication.

## Experiment 1

### Materials and methods

**Ethics statement.** The study was approved by the University of Trento Research Ethics Committee and was conducted following the principles outlined in the Declaration of Helsinki. All volunteers provided written informed consent before starting the experiment and received monetary reimbursement, course credits or a university gadget.

**Participants.** Participants were recruited using the recruitment platform and the social media pages of the University of Trento. The minimally required sample size was N = 34, based on a-priori power analysis in G*Power [75] with a power of 0.8 to detect a medium effect size of Cohen's d = 0.5 at alpha = 0.05. Accordingly, 34 participants were recruited and included in the final analyses (20 females; mean age 24, range 18–47 years). All participants were native speakers of Italian and reported normal or corrected-to-normal vision, normal hearing and no history of neurological or psychiatric conditions.

**Stimuli.** The stimuli consisted of audiovisual recordings of a female speaker uttering Italian words (for the communicative condition) or vocalizing the Italian vowel sound/a/ (for the non-communicative condition). The words were Italian common nouns divided into 22 groups according to their meaning. During the recording, the camera was placed in three different positions along the azimuth for the speaker to be on the right side, left side or at the centre of the screen, at respectively 9°, −9° and 0° of visual angle, to ensure that the speaker's head, gaze, and torso orientation always pointed to the position of the participant (Fig 1a). This was done to give the impression that the speaker was addressing the listener in the communicative condition from each of the three positions. In other words, whether the speaker appeared at −9°, 0°, or +9°, they addressed the listener directly, as is typically the case when starting a conversation in natural settings. This way, the participants would perceive the speaker's movements as directed toward them. The participants were told that the speaker's position would change to make the interaction task more challenging, similar to how communication can be more difficult when someone approaches us to initiate a communicative interaction in a noisy environment, like a busy restaurant. Words were recorded evenly between positions, and the vocalisation was recorded from all three positions. To address the participant in the communicative condition, each video started with the speaker looking down with her head bowed, then she lifted her head vertically, looked at the camera and uttered the word. Conversely, in the non-communicative condition, the speaker kept the head down and then produced a meaningless vocalization [8]. Each synchronous audiovisual stimulus exhibited a natural offset between the audio and video components, with bodily signals preceding the vocalization by approximately 500 milliseconds. Collectively, we introduced relevant linguistic content and bodily signals in the communicative condition to maximize its difference from the non-communicative condition, thereby establishing a clear proof-of-concept for the influence of communicative intent

**a. Stimuli recording setup**

**b. Experimental design**

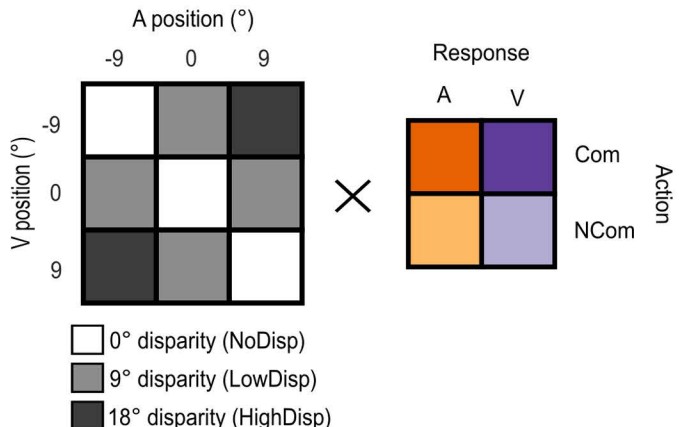

**c. Experimental procedure**

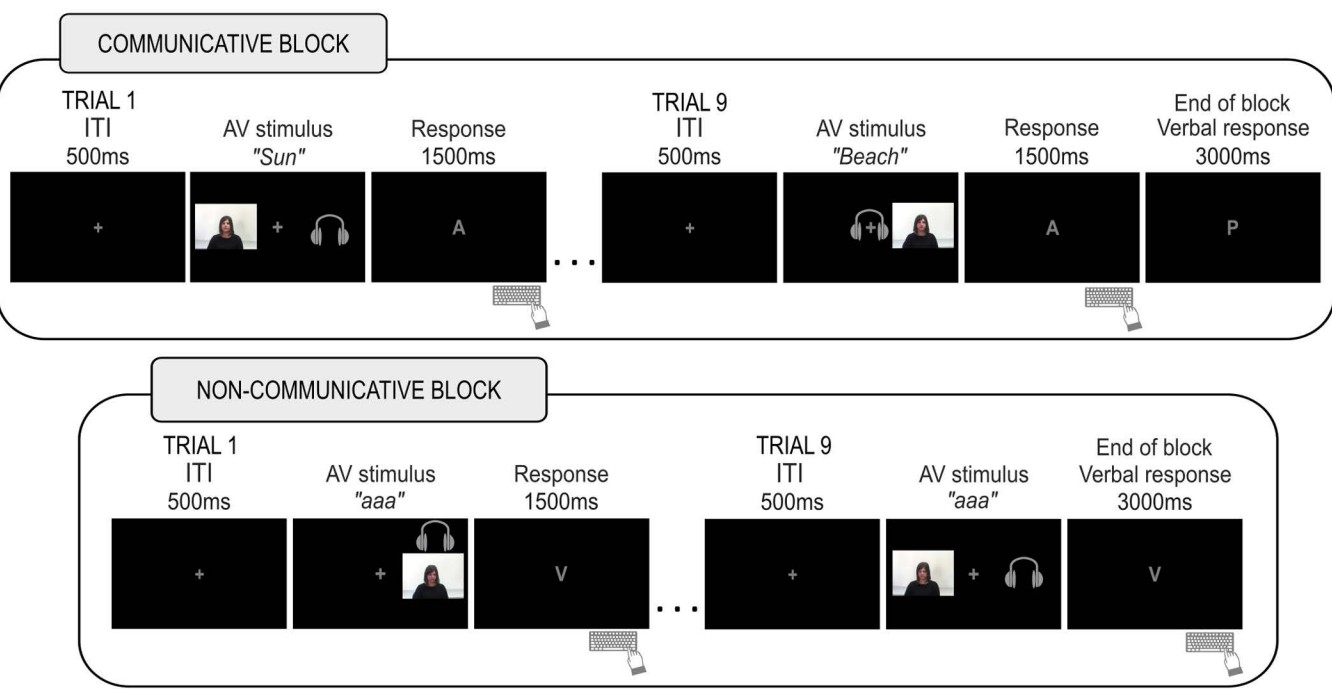

**Fig 1. Stimuli, design and procedure.** a) Visual stimuli were recorded from three different positions. D represents the distance between the speaker and the central camera (D = 150 cm). S represents the horizontal shift of the camera (23.6 cm) to record the speaker at an angle $\alpha$ (9 degrees visual angle), as defined by the formula: **S = 2D tan** $\left(\frac{\alpha}{2}\right)$. In the main experiment, videos with different viewing angles were presented on the side of the screen giving the impression that the speaker was facing the participant (e.g., videos recorded from the right side of the speaker were presented on the left side of the screen). b) We used a 3 (visual stimuli position) × 3 (auditory stimuli position) × 2 (action intention) × 2 (response modality) factorial design, resulting in 36 total experimental conditions. We presented all 9 possible combinations of auditory and visual positions, generating 3 levels of spatial disparity: spatially congruent, i.e., sampled from the same spatial position (0°, white cells); low AV disparity (9°, light grey cells); high AV disparity (18°, dark grey cells). c) The trial started with a 500 ms fixation cross at the centre of the screen, followed by the synchronous audiovisual stimulus. In communicative blocks, the stimuli were videos of the speaker addressing the participant and uttering a word, which lasted on average 1500 ms (range 1000-2000 ms) depending on word length. In non-communicative blocks, the stimuli lasted 500 ms in Experiment 1 and 1500 ms in Experiment 2. After the stimuli offset, a letter on the screen ('A' for auditory and 'V' for visual) cued participants to report the auditory or visual position using a specific keyboard key (1500 ms

response window). At the end of communicative blocks, participants were prompted by the letter 'P' on the screen (for 'parola', i.e., 'word' in Italian) to say a word representing the general theme that connected the words in the block. The experiment consisted of 4 runs, each containing 18 blocks (9 communicative and 9 non-communicative) and 9 trials per block. Please note that the figure exemplifies a few representative audiovisual spatial combinations from all those employed in the study. For each trial, participants saw the visual component on the screen and heard the voice in their headphones. The headphones icon represents the audio location for visualization purposes (it was not presented on the screen during the experiments).

on audiovisual causal inference. While this approach introduces both low- and high-level differences between conditions (such as stimulus dynamics, attentional demands, and semantic content), we view these as intrinsic features of real-world communicative contexts.

For all stimuli, the audio was recorded with a microphone facing the speaker, and background noise was removed afterwards using iMovie (macOS). The auditory stimuli were convolved with spatially selective head-related transfer functions (HRTFs) derived from the ARI HRTF Database [76]. Specifically, stimuli were generated using the *interpolateHRTF* function from the MATLAB Audio Toolbox (documentation: https://it.mathworks.com/help/audio/ref/interpolatehrtf.html). The function utilises a representative subject from the ARI Database and performs spatial interpolation of the Head-Related Impulse Responses (HRIRs) using a Vector Base Amplitude Panning (VBAP) algorithm [77]. For each desired position, the algorithm identifies the three nearest positions in the dataset, and linearly combines the three corresponding HRIRs using the VBAP-based weights. The visual stimuli spanned $14 \times 10$ degrees visual angle and were presented on a black background. Guided by previous research [57,73], we opted for high visual reliability to ensure that observers could perform causal inference and successfully arbitrate between sensory integration and segregation. As a result, our study was optimised to assess causal inference on the observers' reported sound perception.

**Design and procedure.** In a spatial ventriloquist paradigm, we presented synchronous audiovisual stimuli of a speaker uttering a word or producing a meaningless vocalisation. The auditory information (voice) and the visual bodily cues (video) were independently sampled from three positions along the azimuth (-9°, 0° or 9° of visual angles), generating 9 possible spatial combinations and 3 levels of spatial disparity (0°, 9°, 18° of visual angles; Fig 1b). These spatial configurations influence audiovisual integration reliably, as demonstrated by previous work using a highly similar paradigm [73]. Crucially, we manipulated the action intention: in half of the trials, the speaker directed her gaze toward the participant and uttered a word (communicative action); in the other half, a static frame of the speaker looking down was paired with a meaningless vocalisation (non-communicative action). After the presentation of each audiovisual stimulus, participants were instructed to report the position of either the visual or the auditory stimulus by a letter cue at the centre of the screen ('A' for audio and 'V' for video). The experiment was divided into blocks of 9 trials. The action intention (communicative vs. non-communicative) changed with each block, and the report modality (auditory vs. visual) changed every two blocks. Within each block, each spatial combination was presented once in a randomized order. In summary, the experiment conformed to a 3 (visual stimuli positions: -9°; 0°; 9° of visual angles) × 3 (auditory stimuli position: -9°; 0°; 9° of visual angles) × 2 (action intention: communicative; non-communicative) × 2 (response modality: auditory; visual) factorial design, for a total of 36 experimental conditions. Vertical shifts of head and gaze towards the listener contributed to signalling the speaker's intention to communicate; horizontal displacements between the video and audio locations created spatial incongruency, thereby letting us measure the amount of ventriloquist effect. We then asked whether and how the ventriloquist effect was influenced by communicative (relative to non-communicative) audiovisual signals. Importantly, the axis of gaze movement (vertical) was orthogonal to the axis of audiovisual disparity (horizontal), thereby avoiding potential perceptual or attentional biases due to eye movements that may influence spatial localization along the horizontal plane. Each participant completed 4 runs, with 9 trials/block and 18 blocks/run, thus generating 18 trials/condition and 648 trials in total. Each trial started with a 500 ms grey fixation cross in the middle of the screen on a black background, followed by synchronous auditory and visual stimuli. The fixation cross was present for the entire duration

of the stimuli and participants were instructed to fixate it throughout the experiment. The communicative stimuli lasted on average 1500 ms (range 1000–2000 ms) considering the bodily movements preceding the utterance and the duration of the word. The non-communicative stimuli had a fixed duration of 500 ms. This shorter duration was chosen to optimize the length of the experiment for subsequent neuroimaging studies. After the stimuli disappeared, the letter 'A' or 'V' replaced the fixation cross on the screen to indicate the required response modality. This letter remained on the screen for 1500 ms, during which participants had to respond by identifying the location of the target stimuli using a specific keyboard key (left, down and right arrows for left, central and right locations respectively).

To increase ecological validity and ensure that participants focused not only on the spatial position of the stimuli but also on the communicative aspect and the semantic meaning of the words – similar to a real-life face-to-face conversation – they were informed that the person in the videos was interacting with them via a real-time online connection. Moreover, all nine words within each communicative block were connected by a common theme that the speaker aimed to convey to the participant (for example, "sun", "holidays" and "beach" were linked by the theme "summer"). At the end of each communicative block, participants were prompted by a letter cue in the centre of the screen to guess the theme. As a result, the communicative condition differed from the non-communicative condition in terms of attentional demands and semantic content, as is typically the case in real-world contexts.

**Experimental setup.** The experiment was presented with Psychtoolbox (version 3.0.18) running in MATLAB R2021a on a Windows machine (Microsoft 10). Auditory stimuli were presented via headphones (Sannheiser HD580 precision) and visual stimuli were presented on a monitor (1920 × 1089 pixels resolution; 100 Hz frame rate). Participants sat in a dimly lit cubicle in front of the computer monitor at a viewing distance of 80 cm with their heads positioned on a chin rest to minimise movements. To ensure participants maintained central fixation, we monitored and recorded their ocular movements using an eye tracker device (EyeLink 1000, version 4.594, SR Research Ltd.). At the end of each trial, participants responded using the keyboard's left, down or right arrow keys.

**Statistical analyses - overview.** We limited our analyses to trials without missed responses (i.e., no answer within a 1.5-second response time window), premature responses (RT < 100 ms) or response outliers (| RT | > 3 SD from the across-condition median RT of each participant). Only a few trials were discarded (5.7% ± 0.6% [across participants mean ± SEM]). Further, we excluded trials without central fixation during stimuli presentation. Significant saccades were defined as eye movements with an amplitude greater than 7° (half the width of the video stimuli, hence exceeding the video boundaries). Participants successfully maintained central fixation with only a few rejected trials (3.78% ± 0.42% [across participants mean ± SEM]).

In the following, we describe the main analyses: audiovisual weight index ($w_{AV}$) and Bayesian modelling. Although not central to the present study, we additionally examined response times (see S1 Text).

## Audiovisual weight index *wAV*

To evaluate the effect of auditory and visual signal locations on observers' reported auditory (or visual) locations, we calculated an audiovisual weight index ($w_{AV}$) for the trials where the auditory and visual signals were spatially incongruent (i.e., AV spatial disparity greater than 0). The $w_{AV}$ index is defined as the difference between the reported location in an incongruent trial (e.g., when the visual stimulus is at 0° and the auditory stimulus at 9°) and the average reported location in audiovisual congruent trials (e.g., when audiovisual stimuli are at 9°), scaled by the difference between the average reported locations in the two respective congruent condition (e.g., when audiovisual stimuli are at 0°, and when audiovisual stimuli are at 9°) [73]:

$$w_{AV,XY} = \frac{Reported\ location_{Incongruent,A=X,\ V=Y} - Reported\ location_{Congruent,\ A=V=X}}{Reported\ location_{Congruent,\ A=V=Y} - Reported\ location_{Congruent,\ A=V=X}} \tag{1}$$

The denominator acts as a scaling operator for all conditions at a particular level of spatial disparity; to increase its estimation precision, we computed it using the average reported location in audiovisual congruent trials pooled over all conditions across all participants (A=V=-9°: -8.589°; A=V=0°: -0.667; A=V=9°: 8.208°). Notice that using the perceived locations of the congruent conditions instead of the true stimuli locations accommodates participants' response biases. Under the assumption of comparable biases across all conditions, the $w_{AV}$ index represents the relative influence of the auditory and visual positions on the reported position: a value of 0 indicates that the observer's report relies completely on the auditory signal position; a value of 1 indicates that the observer's report relies completely on the visual signal position. The $w_{AV}$ conceptually corresponds to what is often referred to as crossmodal bias in the literature.

We computed the $w_{AV}$ index for each participant and experimental trial using Eq. 1 and we then averaged the values across all AV combinations at a certain level of spatial disparity (9°: low; 18°: high). Hence, we analysed the $w_{AV}$ index in a 2 (response modality: auditory; visual) × 2 (spatial disparity: low; high) × 2 (action intention: communicative; non-communicative) factorial design. To avoid making parametric assumptions, we evaluated the main effects of response modality, spatial disparity, action intention and their interactions using non-parametric permutation tests at the between-subject level. The observed test statistic was the across-participants' difference in means between conditions (or condition combinations for interaction effects). To construct the null distribution, we randomly permuted the condition labels across participants, preserving the number of participants per condition, and recalculated the test statistic for each permutation. This process was repeated 4,096 times to generate an empirical null distribution. Two-tailed p-values were computed by comparing the observed test statistic to the null distribution, calculating the proportion of permuted statistics that exceeded the observed value. For effect sizes, we report the difference between the across-participants' mean empirical effect and the mean of the nonparametric null distribution and 95% confidence intervals (C.I.) [73].

**General Bayesian modelling approach.** Combining psychophysics and Bayesian modelling, we characterized the computational principles by which the observer integrates multisensory signals during face-to-face communication. In the following, we first describe the Bayesian Causal Inference model from which we derive the Forced Fusion model as a special case (see [52] for further details). Second, we describe how action intention may affect multisensory processing.

The Bayesian Causal Inference (BCI) model formalizes how an ideal observer should combine information deriving from different sensory modalities while considering the uncertainty about the underlying causal structure, i.e., whether sensory events are generated by a common cause or separate causes (Fig 2a).

Briefly, the BCI model assumes that the common ($C=1$) and independent ($C=2$) causes are sampled from a binomial distribution defined by the common cause prior $p_{common}$, which represents the prior probability that the multisensory signals belong to the same source [52]. In other words, $p_{common}$ quantifies the observer's unity assumption, that is the prior belief that the signals should be integrated into one unified percept [37–42]. Such common cause prior is likely supported by observers' expectations [37,39], which in turn may depend on prior experience [43–46], training [48,49], ontogenetic [50] and phylogenetic [51] factors. In the context of spatial localization, we know that auditory and visual stimuli that are spatially congruent tend to originate from the same source. Accordingly, audiovisual spatial disparity impacts the observer's unity assumption: integration increasingly breaks down at higher disparity levels, where auditory and visual signals are less likely to originate from a common source. The computational architecture of the BCI model directly accounts for this effect [52].

Under the assumption of a common cause, the position of the source $S$ is drawn from the prior distribution $N(0, \sigma_P)$, thereby assuming a central bias [57,78–80] with a strength represented by $\sigma_P$. Under the assumption of separate causes, the positions $S_V$ and $S_A$ are drawn independently from the same distribution $N(0, \sigma_P)$. We then assume that auditory and visual signals are corrupted by independent Gaussian noise of standard deviations $\sigma_V$ and $\sigma_A$; thus, the events $x_V$ and $x_A$ are drawn respectively from the independent Gaussian distributions $N(S_V, \sigma_V)$ and $N(S_A, \sigma_A)$ [52]. Accordingly, the generative model employs four free parameters: common cause prior probability ($p_{common}$), standard deviation of the spatial prior ($\sigma_P$), and standard deviation of the auditory and visual representations ($\sigma_A$ and $\sigma_V$).

**a. Generative models**

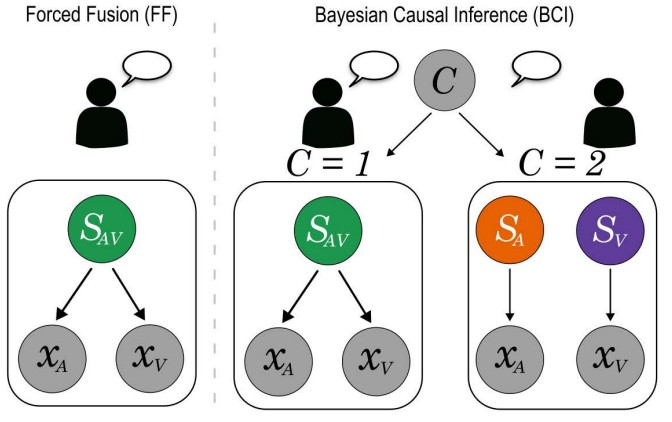

**b. Factorial model space**

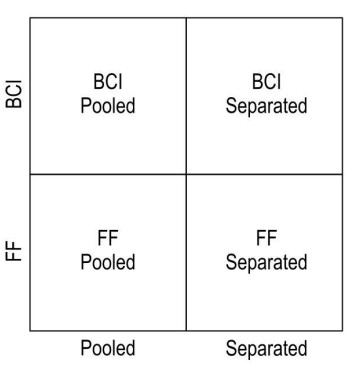

**Fig 2. Bayesian Modelling.** a) Generative models of Forced Fusion (FF) and Bayesian Causal Inference (BCI). For FF, a single source mandatorily generates auditory and visual signals. Instead, BCI explicitly models the two causal structures, i.e., whether auditory and visual signals come from one common cause ($C=1$) or separate causes ($C=2$). The FF model assumes that the auditory ($\mathbf{x}_A$) and the visual ($\mathbf{x}_V$) signals originate from the same source ($\mathbf{S}_{AV}$). The BCI model accounts for causal uncertainty, i.e., the possibility that the auditory ($\mathbf{x}_A$) and the visual ($\mathbf{x}_V$) signals could originate from one common ($C=1$) or two independent ($C=2$) causes. Due to its model architecture (see section "General Bayesian modelling approach"), only BCI models accommodate spatial disparity and response modality effects. b) To determine whether a BCI or FF model best explained each participant's localisation responses, and to evaluate the modulatory influence of action intention (communicative vs. non-communicative), we performed Bayesian model comparison in a 2 (BCI vs FF) × 2 (Pooled vs Separated action intention conditions) factorial model space.

The posterior probability of the underlying causal structure is inferred by combining the common cause prior $p_{\mathrm{common}}$ with the sensory evidence according to Bayes' rule:

$$p\left(C \mid x_V, \ x_A\right) = \frac{p\left(x_V, \ x_A \mid C\right) p(C)}{p(x_V, \ x_A)} \tag{2}$$

Depending on the causal structure, the brain generates the respective perceptual estimates by linearly weighting the signals $x_V$ and $x_A$ by the reliability of the respective sensory modality, i.e. the inverse of their variances. In the case of two independent sources, the optimal visual and auditory estimates are:

$$\hat{S}_{V,C=2} = \frac{\frac{x_V}{\sigma^2_V} + \frac{\mu_P}{\sigma^2_P}}{\frac{1}{\sigma^2_V} + \frac{1}{\sigma^2_P}} \ ; \tag{3}$$

$$\hat{S}_{A,C=2} = \frac{\frac{x_A}{\sigma^2_A} + \frac{\mu_P}{\sigma^2_P}}{\frac{1}{\sigma^2_A} + \frac{1}{\sigma^2_P}} \tag{4}$$

In the case of a common cause, thus $\hat{S}_V = \hat{S}_A$, the optimal estimate is:

$$\hat{S}_{V,C=1} = \hat{S}_{A,C=1} = \frac{\frac{x_V}{\sigma^2_V} + \frac{x_A}{\sigma^2_A} + \frac{\mu_P}{\sigma^2_P}}{\frac{1}{\sigma^2_V} + \frac{1}{\sigma^2_A} + \frac{1}{\sigma^2_P}} \tag{5}$$

To compute the final estimate of the auditory and visual locations under causal uncertainty, the brain may use various decision functions, such as "model averaging", "model selection" and "probability matching" [81]. In line with previous

work [58,59,73,82], we focused on model averaging, which integrates the estimates of each causal structure weighted by the posterior probability of the respective structure. Accordingly, the final perceptual estimates of the auditory and visual source locations are computed as:

$$\hat{S}_V = p(x_V, x_A)\,\hat{S}_{V,C=1} + (1 - p(x_V, x_A))\hat{S}_{V,C=2} \tag{6}$$

$$\hat{S}_A = p(x_V, x_A)\,\hat{S}_{A,C=1} + (1 - p(x_V, x_A))\hat{S}_{A,C=2} \tag{7}$$

Thus, the Bayesian Causal Inference model relies on three spatial estimates, $\hat{S}_{AV,C=1}$, $\hat{S}_{V,C=2}$ and $\hat{S}_{A,C=2}$, which are combined into one final estimate according to task relevance ($\hat{S}_A$ for auditory response, $\hat{S}_V$ for visual response). In other words, the final perceptual estimate reflects the flexible readout of internal estimates according to current task demands [73]. Collectively, two effects naturally arise from the BCI model architecture: first, integration weakens when auditory and visual signals are sampled from more distant locations (effect of spatial disparity); second, task relevance dictates which internal perceptual estimates are combined to generate the final perceptual response (effect of response modality).

The Forced Fusion (FF) model can be considered a particular scenario of the Bayesian Causal Inference model, in which it is assumed with certainty that the audiovisual signals come from one common source and are therefore mandatorily integrated (i.e., $p_{common}$ = 1). The final spatial audiovisual estimate $\hat{S}_{AV}$ is directly computed as a reliability-weighted linear average of the two unisensory estimates as described by Eq. 5 [83]. Thus, the FF model includes only three free parameters: standard deviation of the spatial prior ($\sigma_P$), and standard deviation of the auditory and visual representations ($\sigma_A$ and $\sigma_V$). As a result, the FF model architecture does not accommodate the effects of spatial disparity or response modality.

To determine whether a BCI or FF model would best explain each participant's localisation responses, and to evaluate the modulatory influence of action intention (communicative vs. non-communicative), we performed Bayesian model comparison in a 2 (BCI vs FF architecture) × 2 (Pooled vs Separated action intention conditions) factorial model space (Fig 2b). Thus, we compared 4 models: (i) a BCI model that does not account for action intention, (ii) a BCI model with separate parameters for each action intention condition, (iii) an FF model that does not account for action intention, and (iv) an FF model with separate parameters for each action intention condition. By comparing FF vs BCI models, we evaluated whether observers always integrate auditory and visual stimuli regardless of causal uncertainty (in line with FF), or if they combine the two alternative perceptual estimates weighted by the posterior probabilities of the respective causal structure (in line with BCI) as described by Eq. 6 and 7. This would confirm that prior expectations arbitrate between the integration and segregation of vocal and bodily signals during face-to-face communication [4,5]. By comparing Pooled vs Separated models, we evaluated whether observers showed a stronger tendency to integrate vocal and bodily information for communicative than non-communicative signals. Crucially, by comparing the BCI versus FF Separated models we characterized the exact computational mechanism by which communicative signals may increase multisensory integration. On the one hand, the perceptual characteristics of the audiovisual stimuli may influence the reliabilities of the sensory information before integration in both the FF and the BCI models. Specifically, the speaker may capture the observer's attention more effectively by addressing them with head and gaze (communicative condition) as opposed to keeping the head down (non-communicative condition), thereby decreasing the noise (i.e. increase the reliability) of the attended visual information [73]. This would be captured by a decrease of $\sigma_V$ in the communicative than non-communicative condition, which would increase the visual weight during the fusion process under the communicative condition. Alternatively, the shared communicative nature of the video and audio stimuli may reinforce the prior belief that the two signals come from a common cause, as captured by an increase of $p_{common}$ in the BCI model for communicative than non-communicative trials. The factorial model comparison allowed us to directly arbitrate between these competing hypotheses.

To summarize, the BCI Pooled model employs four free parameters ($p_{common}$, $\sigma_P$, $\sigma_V$ and $\sigma_A$), which are doubled in the BCI Separated model (communicative condition: $p_{common\ Com}$, $\sigma_{P\ Com}$, $\sigma_{V\ Com}$ and $\sigma_{A\ Com}$; non-communicative condition: $p_{common\ NCom}$, $\sigma_{P\ NCom}$, $\sigma_{V\ NCom}$ and $\sigma_{A\ NCom}$). The FF Pooled model includes three free parameters ($\sigma_P$, $\sigma_V$ and $\sigma_A$). Following the same logic as above, these three parameters are doubled in the FF Separated model that considers the two action intentions separately (communicative condition: $\sigma_{P\ Com}$, $\sigma_{V\ Com}$ and $\sigma_{A\ Com}$; non-communicative condition: $\sigma_{P\ NCom}$, $\sigma_{V\ NCom}$ and $\sigma_{A\ NCom}$).

We fitted each model to the individual participants' localisation responses based on the predicted distributions of the spatial estimates. The distributions were obtained by marginalising over the internal variables not accessible to the experimenter ($x_V$ and $x_A$), which were simulated 10,000 times for each of the 36 conditions (see 'Experimental Design'), and inferring the spatial estimates from Eqs. 2–7 for each simulated $x_V$ and $x_A$. Thus, we obtained a histogram of the auditory and visual responses predicted by each model for each condition and participant; based on these, we applied the multinomial distribution to compute the probability of each participant's counts of localisation responses [73]. This gave us the likelihood of each model given the participants' responses. Assuming independence of conditions, we summed the log-likelihoods across conditions. To obtain maximum likelihood estimates for the parameters of the models, we used a nonlinear simplex optimisation algorithm as implemented in MATLAB's *fminsearch* function (MATLAB R2022b), after initialising the parameters in a prior grid search.

We firstly assessed the model fit for the data using the coefficient of determination $R^2$, defined as:

$$R^2 = 1 - exp - \frac{2}{n}(I\left(\hat{\beta}\right) - I(0))$$

(8)

where $I(0)$ and $I\left(\hat{\beta}\right)$ represent respectively the log likelihood of the null model and the fitted model, and $n$ is the number of data points. As the null model, we assumed a random response between the three possibilities (i.e. left, centre, right), hence a discrete distribution with a probability of 0.33. As in our case the models' responses were discretized to relate them to the three discrete response options, and the coefficient of determination was scaled (i.e., divided) by the maximum coefficient defined as:

$$max\left(R^2\right) = 1 - exp(\frac{2}{n}I(0))$$

(9)

To identify the model that fits the localisation responses best, we compared the models using the Bayesian Information Criterion as an approximation to the log model evidence [84]:

$$BIC = -ln\hat{L} + k \times ln\ n$$

(10)

where $\hat{L}$ represents the likelihood, $k$ the number of parameters and $n$ the number of data points. We then used SPM12 software (Statistical Parametric Mapping, Functional Imaging Laboratory, UCL) to perform random-effect Bayesian model selection at the group level [85]. This allowed us to estimate the protected exceedance probability that one model explains the data better than any other models and above the chance level, as quantified by the Bayesian Omnibus Risk (BOR). Lastly, for the winning model (i.e. the BCI Separated model), we evaluated pairwise parameter changes as a function of action intention (communicative vs. non-communicative). To refrain from parametric assumptions, we employed non-parametric permutation tests at the between-subject level (same statistical procedure as for the $w_{AV}$ index). For effect sizes, we report the difference between the across-participants' mean empirical effect and the mean of the nonparametric null distribution and 95% CI [73]. Finally, we supplement these analyses with Bayes factors ($BF_{10}$) from Bayesian non-parametric pairwise tests implemented in JASP 0.19.1.

## Results

**Audiovisual weight index *wAV*.** To evaluate *whether* response modality, spatial disparity and action intention influenced multisensory integration, we examined their main effects and interactions on the audiovisual weight index $w_{AV}$, which quantifies the impact of the visual and auditory signals' positions on participants' spatial reports. Results are shown in Fig 3 (top panels) and summarized in Table 1 (see also S1 Table for comprehensive descriptive statistics). We found a significant main effect of response modality ($p < 0.001$, es = 0.83, CI(95%) = [0.79, 0.86]), indicating that participants relied more on the auditory signal when reporting the auditory position, and relied more on the visual signal when

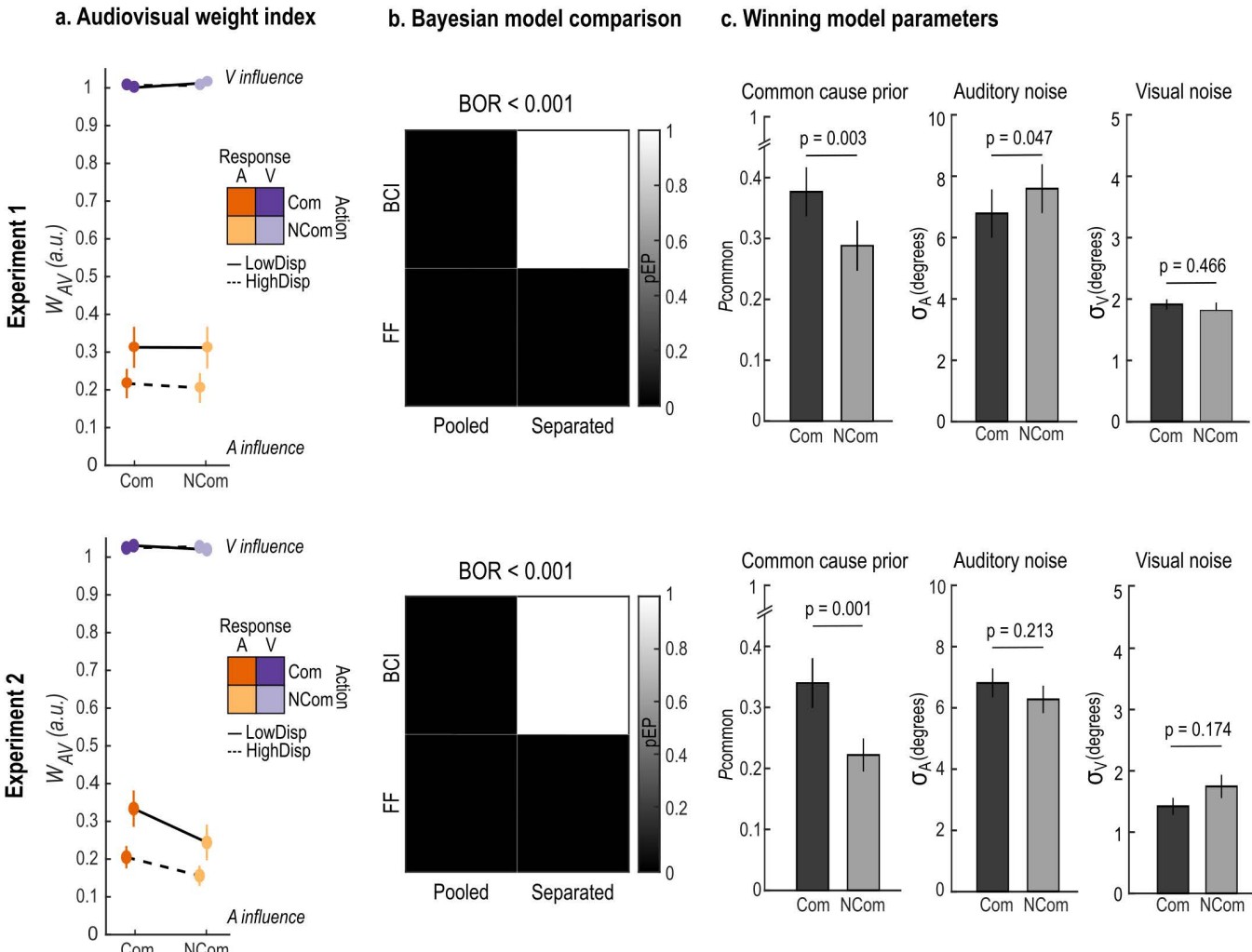

**Fig 3. Audiovisual weight index and modelling results.** a) Across participants' mean ± SEM audiovisual weight index ($w_{AV}$) as a function of action intention (communicative: Com; non-communicative: NCom), response modality (auditory: A; visual: V) and audiovisual spatial disparity (9°: LowDisp; 18°: HighDisp). An index equal to 1 (respectively, 0) indicates pure visual (respectively, auditory) influence on participants' localization responses; values between 0 and 1 indicate intermediate degrees of audiovisual integration. b) Protected exceedance probability (pEP, grayscale) of each model in the 2×2 model comparison factorial space (see Fig 2b), i.e. how likely each model is to explain the data compared to the other models. Bayesian Omnibus Risk (BOR) represents the probability that the results are due to chance. c) Across participants' mean ± SEM parameter estimates of common cause prior ($p_{common}$), auditory noise ($\sigma_A$), and visual noise ($\sigma_V$) of the winning model (i.e., BCI Separated) as a function of action intention. p-values based on two-tailed between-subject permutation tests. Top and bottom rows show the results of Experiment 1 and 2 respectively.

**Table 1. Audiovisual weight index (*wAV*): statistical significance.** Main effects and interactions for the audiovisual weight index (*wAV*) in the 2 (action intention: communicative; non-communicative) × 2 (response modality: auditory; visual) × 2 (spatial disparity: low; high) factorial design. P-values are based on two-tailed between-subjects permutation tests. Effect sizes [95% CI] correspond to the difference between the across-participants' mean empirical effect and the mean of the non-parametric null distribution.

| | Experiment 1 | | | Experiment 2 | | |
|---|---|---|---|---|---|---|
| | p-value | Effect size | 95% C.I. | p-value | Effect size | 95% C.I. |
| Act | 1 | -0.01 | [-0.03, 0.00] | **0.035** | 0.03 | [0.01, 0.05] |
| Resp | **< 0.001** | 0.83 | [0.79, 0.86] | **< 0.001** | 0.81 | [0.77, 0.85] |
| Disp | **< 0.001** | -0.04 | [-0.05, -0.03] | **< 0.001** | -0.05 | [-0.06, -0.03] |
| Act×Resp | 0.577 | -0.00 | [-0.03, 0.02] | **0.047** | -0.05 | [-0.09, -0.02] |
| Act×Disp | 0.525 | 0.00 | [-0.02, 0.02] | 0.148 | -0.02 | [-0.04, 0.00] |
| Resp×Disp | **< 0.001** | 0.09 | [0.08, 0.11] | **< 0.001** | 0.07 | [0.04, 0.09] |
| Act×Resp×Disp | 0.951 | 0.03 | [-0.00, 0.06] | 0.483 | -0.03 | [-0.07, 0.01] |

reporting the visual position (Fig 3a). Thus, task relevance influenced participants' localization responses, corroborating the presence of a flexible readout mechanism of internal perceptual estimates according to current task demands [58,73]. Additionally, we found a significant response modality × spatial disparity interaction ($p < 0.001$, es = 0.09, CI(95%) = [0.08, 0.11]). Post hoc permutation testing showed that the $w_{AV}$ index depended on spatial disparity when participants reported the auditory position ($p < 0.001$, es = -0.09, CI(95%) = [-0.11, -0.07]), reflecting a greater influence of the visual stimuli on spatial localization when audiovisual signals were closer in space. These results confirm that integrating signals from different sensory modalities does not conform to Forced Fusion principles [60–62,86–88]: observers did not mandatorily integrate sensory signals weighted by their sensory reliabilities into one unified percept, hence reporting the same location irrespective of the task context. Instead, task relevance and spatial disparity modulated the extent to which a visual signal influences the perception of a sound's position [58,59,73,89]. Importantly, response biases may vary across participants and experimental conditions. To account for these individual biases more stringently, we recomputed the $w_{AV}$ using the average reported location in audiovisual congruent trials for each participant and each of the four (action intention × response modality) experimental conditions. This sensitivity analysis (S3 Fig and S8 and S9 Tables) yielded results consistent with the original analysis, confirming the robustness of our findings.

Critically, we did not find an effect of action intention on the $w_{AV}$ index, suggesting that the relative strength of the two sensory modalities on participants' perceived target location was indistinguishable across communicative and non-communicative stimuli. To understand why that is, we turn to the modelling results.

**Bayesian modelling.** To evaluate *how* response modality, spatial disparity and action intention influenced multisensory integration, we formally characterized the computational principles underlying participants' localization responses. We compared 4 models of audiovisual spatial integration in a 2 (BCI vs. FF model architecture) × 2 (Pooled vs. Separated action intention conditions) factorial model space. Specifically, we contrasted (i) a Bayesian Causal Inference model with pooled parameters across action intention conditions, (ii) a Bayesian Causal Inference model with separate free parameters for each action intention condition, and similarly (iii) a Force Fusion model with pooled parameters and (iv) a Forced Fusion model with separate free parameters. The model comparison analysis provided overwhelming evidence in favour of the BCI Separated model (variance explained $R^2 = 0.94 \pm 0.01$, protected exceedance probability ≈ 1, Bayesian Omnibus Risk < 0.001; Fig 3b; see also S3 Table). This result indicates that response modality, spatial disparity, and, crucially, also action intention influenced audiovisual spatial localization, since the BCI Separated model is the only one that accommodates all effects. Specifically, BCI captures response modality and spatial disparity effects due to its model architecture; further, the Separated model fitting procedure captures any modulating effects of action intention. To explicitly characterize these effects, we performed pairwise comparisons of the model's parameters

between the two action intention conditions (Fig 3c). Results showed that the common cause prior $p_{common}$ was significantly higher in the communicative relative to non-communicative condition ($p = 0.003$, es $= 0.08$, CI(95%) = [0.02, 0.15], $BF_{10} = 155.6$) suggesting that observers held a greater a priori expectation that audiovisual signals originate from a common cause in a communicative context. Instead, the visual noise $\sigma_V$ did not differ across the two action intention conditions ($p = 0.466$, es $= 0.18$, CI(95%) = [-0.09, 0.44], $BF_{10} = 0.49$). Hence, the perceptual characteristics of the visual stimuli (communicative: dynamic video of speaker addressing the observer; non-communicative: static frame of speaker looking down) did not modulate visual reliability and thus did not change the weight assigned to the visual modality during audiovisual integration across the two conditions. Critically, however, the auditory noise $\sigma_A$ was significantly higher in the non-communicative condition ($p = 0.047$, es $= -0.85$, CI(95%) = [-1.61, -0.10], $BF_{10} = 10.3$), indicating that the auditory information was more noisy (i.e., less reliable) and thereby carried less weight during the fusion process (for comprehensive results, see S2 Table).

To formally test the functional connection between the model parameters and the $w_{AV}$ we performed a follow-up correlation analysis (Fig 4). In line with the principles of BCI, we found that individual increases of $w_{AV}$ for the communicative relative to the non-communicative condition ($w_{AV,Com} > w_{AV,NCom}$) positively correlated with the respective effect on the causal prior ($p_{common,Com} > p_{common,NCom}$) and auditory noise ($\sigma_{A,Com} > \sigma_{A,NCom}$). Both correlations were highly significant ($p_{common}$: $r = 0.78$, $p < 0.001$; $\sigma A$: $r = 0.69$, $p < 0.001$). Together, it is then plausible that two opposing forces across the two action intention conditions interacted to shape the spatial perceptual estimates, ultimately neutralizing any effects on the final response (and, thus, on the $w_{AV}$ index). On the one hand, multisensory integration was enhanced in the communicative condition through a stronger common cause prior. On the other hand, the visual capture of sound was enhanced in the non-communicative condition due to higher auditory noise. A decrease in auditory reliability decreased the weight of the auditory information and thereby its influence on the perceived sound position, thereby determining a stronger visual capture. Since auditory localization accuracy decreases for shorter sounds [90,91], we reasoned that the increase of auditory noise in the non-communicative condition likely stemmed from the shorter stimuli duration relative to the communicative condition. Therefore, equalizing stimulus duration across the two action intention conditions should eliminate this effect, without affecting the common cause prior. If so, we would expect a stronger visual capture in the communicative condition through a higher common cause prior. We directly tested this hypothesis in Experiment 2.

## Experiment 2

### Materials and methods

**Participants.** Participants were recruited using the recruitment platform and the social media pages of the University of Trento. The minimally required sample size was N = 34, based on a-priori power analysis in G*Power [75] with a power of 0.8 to detect a medium effect size of Cohen's d = 0.5 at alpha = 0.05. Importantly, we included a unisensory auditory screening to ensure participants could distinguish and correctly localize the auditory stimuli without visual interference. This way, any effects in the main experiment could be more safely interpreted in terms of visual capture of perceived sound location, instead of a response that arises from perceptual confusion or uncertainty. Observers located auditory signals – the same auditory stimuli presented during the experiment – randomly presented at −9˚, 0˚, or 9˚ visual angle along the azimuth. Their auditory localization accuracy was quantified by the root mean square error (RMSE) between the participants' reported location and the signal's true location (low RMSE indicates high accuracy). Based on similar research [73], observers were excluded as outliers if their RMSE was greater than 6˚. We recruited 41 volunteers, 7 of which were excluded due to poor auditory localization. As a result, the final analyses included 34 participants (26 females; mean age 23, range 18–47 years), who demonstrated reliable auditory localization (proportion correct responses = 0.81 ± 0.02, mean ± SEM; Pearson's correlation coefficient between true and reported location: r = 0.89 [0.86, 0.92], mean ± 95% CI). All participants were native speakers of Italian and reported normal or corrected-to-normal vision, normal hearing and no history of neurological or psychiatric conditions.

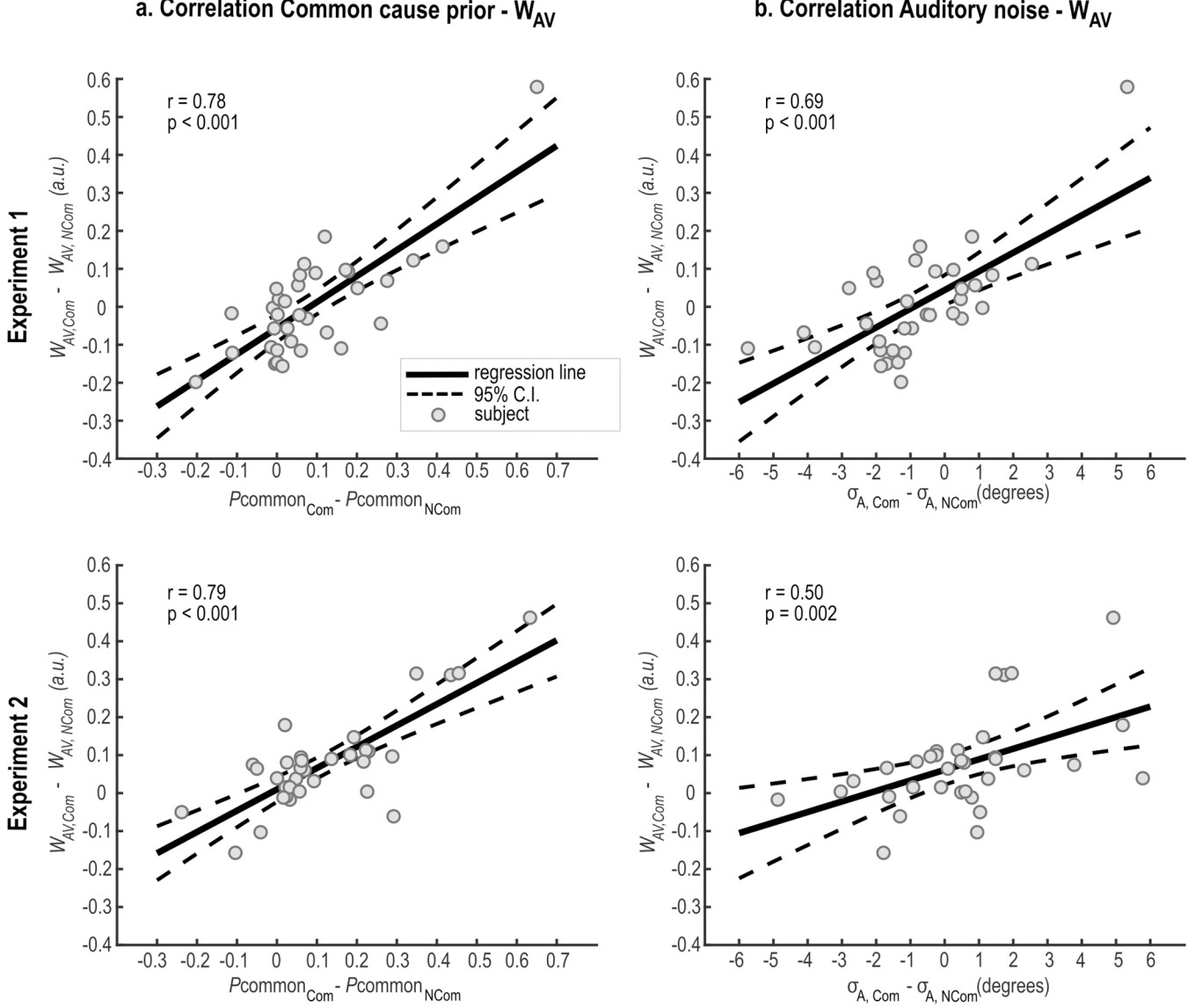

**Fig 4. Correlation between audiovisual weight index (*wAV*) and BCI model parameters.** Across participants' correlation between increases of $w_{AV}$ for the communicative relative to the non-communicative condition ($\mathbf{w}_{AV,Com} - w_{AV,NCom}$) and the respective effect on **a)** the common cause prior ($\mathbf{p}_{common,Com} - p_{common,NCom}$) and **b)** the auditory noise ($\sigma_{A,Com} - \sigma_{A,NCom}$) of the winning model (BCI Separated). Pearson's correlation coefficients (r) and p-values (p) obtained from the two linear regression models: (1) $w_{AV} = \beta_0 + \beta_1 \times p_{common} + \varepsilon$ and (2) $w_{AV} = \beta_0 + \beta_1 \times \sigma_A + \varepsilon$, where $\beta_0$ = intercept, $\beta_1$ = slope coefficient, $\varepsilon$ = residual error. Thick black line and dashed lines respectively show the linear regression line and the 95% confidence intervals (C.I.) from the linear regression models. Grey dots represent individual participants. Top and bottom rows show the results of Experiments 1 and 2 respectively.

**Ethics statement.** The study was approved by the University of Trento Research Ethics Committee and was conducted following the principles outlined in the Declaration of Helsinki. All volunteers provided written informed consent before starting the experiment and received monetary reimbursement, course credits or a university gadget.

**Stimuli, design and procedure.** The design and procedure of Experiment 2 were identical to those of Experiment 1 in every respect (Fig 1), except for the duration of the non-communicative trials. In Experiment 1, non-communicative stimuli had a fixed duration of 500 ms, which was significantly shorter than the average duration of communicative stimuli

(1500 ms), including the bodily movements preceding the utterance and the word itself. In Experiment 2, the duration of the non-communicative trials was extended to 1500 ms to match the communicative trials, thereby eliminating any potential confounds caused by differing stimulus durations.

**Statistical analyses - overview.** We followed the same analytical rationale as in Experiment 1. Briefly, we limited our analyses to trials without missed responses (i.e., no answer within a 1.5-second response time window), premature responses (RT < 100 ms) or response outliers (| RT | > 3 SD from the across-condition median RT of each participant). Only a few trials were discarded (4.3% ± 0.5% [across participants mean ± SEM]). Further, we excluded trials without central fixation during stimuli presentation. Significant saccades were defined as eye movements with an amplitude greater than 7° (half the width of the video stimuli, hence exceeding the video boundaries). Participants successfully maintained central fixation with only a few rejected trials (6.28% ± 1.0% [across participants mean ± SEM]). We then conducted the same analyses as in Experiment 1 (audiovisual weight index $w_{AV}$ and Bayesian modelling). For the $w_{AV}$ index, we followed Eq. 1 and applied the same correction as in Experiment 1. Specifically, we computed the denominator using the average reported location in audiovisual congruent trials pooled over all conditions across all participants (A = V = -9°: -8.668°; A = V = 0°: -0.639; A = V = 9°: 7.849°). For Bayesian modelling, we again performed Bayesian model comparison in the 2 (BCI vs FF architecture) × 2 (Pooled vs Separated action intention conditions) factorial model space (Fig 2b). For the winning model (i.e., the BCI Separated model), we evaluated pairwise parameter changes as a function of action intention (communicative vs. non-communicative). Although not central to the present study, we additionally examined response times (see S1 Text).

## Results

**Audiovisual weight index *wAV*.** Results are shown in Fig 3 (bottom panels) and summarized in Table 1 (see also S1 Table for comprehensive descriptive statistics). We confirmed the significant main effect of response modality ($p < 0.001$, es = 0.81, CI(95%) = [0.77, 0.85]): the influence of the auditory (respectively, visual) signals on participants' localization increased for auditory (respectively, visual) responses. Further, we confirmed the significant response modality × spatial disparity interaction ($p < 0.001$, es = 0.07, CI(95%) = [0.04, 0.09]) (Fig 3a): the influence of the visual stimulus on auditory localization increased at lower than higher spatial disparities ($p < 0.001$, es = -0.07, CI(95%) = [-0.10, -0.06]). Crucially, we also found a main effect of action intention ($p = 0.035$, es = 0.03, CI(95%) = [0.01, 0.05]) (Table 1): audiovisual integration increased for the communicative compared to the non-communicative condition. In particular, we found a significant action intention × response modality interaction ($p = 0.047$, es = -0.05, CI(95%) = [-0.09, -0.02]): the influence of the visual stimulus on auditory localization was greater in the communicative condition (p = 0.036, es = -0.06, CI(95%) = [-0.10, -0.02]. Moreover, a sensitivity analysis using the average reported location in audiovisual congruent trials for each participant and each of the four (action intention × response modality) experimental conditions confirmed the robustness of this effect (S3 Fig and S8 and S9 Tables). We expected these findings to originate from a higher common cause prior in the case of communicative stimuli. To evaluate this hypothesis, we turn to the modelling results.

**Bayesian modelling.** Confirming the results from Experiment 1, the model comparison analysis provided again overwhelming evidence in favour of the BCI Separated model (variance explained $R^2 = 0.94 ± 0.01$, protected exceedance probability ≈ 1, Bayesian Omnibus Risk < 0.001) (Fig 3b; see also S3 Table). Further, we also confirmed that the common cause prior $p_{common}$ increased in the communicative relative to the non-communicative condition ($p = 0.001$, es = 0.11, CI(95%) = [0.04, 0.18], $BF_{10} = 259.6$) (Fig 3c). Conversely, we found no significant difference between the two action intention conditions for the auditory noise $\sigma_A$ (p = 0.213, es = 0.47, CI(95%) = [-0.36, 1.29], $BF_{10}$ = 0.58) and visual noise $\sigma_V$ (p = 0.174, es = -0.46, CI(95%) = [-0.89, -0.03], $BF_{10}$ = 0.85) parameters (for comprehensive results, see S2 Table). Again, we performed a follow-up correlation analysis between the model parameters and the $w_{AV}$ (Fig 4). Following the principles of BCI, we found that individual increases of $w_{AV}$ for the communicative relative to the non-communicative condition ($w_{AV,Com} > w_{AV,NCom}$) positively correlated with the respective effect on the causal prior ($p_{common,Com} > p_{common,NCom}$) and auditory noise (($\sigma_{A,Com} > \sigma_{A,NCom}$). Both correlations were highly significant ($p_{common}$: r = 0.79, $p < 0.001$; $\sigma_A$: r = 0.50, $p = 0.002$). Together, these findings

corroborate the hypothesis that two contrasting forces were at play in Experiment 1. On the one hand, multisensory integration was enhanced in the communicative condition through a higher common cause prior. On the other hand, the visual capture of sound was enhanced in the non-communicative condition due to higher auditory noise. We eliminated this effect in Experiment 2 by matching the stimuli duration across the two action intention conditions. We thereby revealed that a greater prior tendency to integrate vocal and bodily information for communicative than non-communicative signals directly influences audiovisual integration as expressed by participants' behaviour (and, thus, the $w_{AV}$ index).

## Assessing non-optimal decision strategies

In our previous analyses, we employed a normative Bayesian framework in which an ideal observer optimally performs perceptual inference. Critically, Forced Fusion offers a relatively weak alternative to Bayesian Causal Inference, as the rigid common-cause assumption is unlikely to hold in the presence of varying spatial disparities. However, models implementing non-optimal decision strategies—such as Fixed Criterion and Stochastic Fusion [55]—offer psychologically plausible, heuristic approximations to perceptual inference that can better accommodate scenarios where observers do not always integrate the signals. Thus, we conducted a follow-up analysis expanding our model space to include Fixed Criterion and Stochastic Fusion models as plausible and stronger model competitors to Bayesian Causal Inference. In the following, we describe the models' architecture, outline the model comparison approach, and summarize the key findings.

**Bayesian modelling.** The Fixed Criterion and Stochastic Fusion models are equal to Bayesian Causal Inference (see section "General Bayesian modelling approach") except for how they arbitrate between multisensory integration and segregation.

The Fixed Criterion (FC) model exhibits sensitivity to spatial disparity by applying a static disparity threshold to decide whether to integrate or segregate, disregarding the relative sensory uncertainties. Formally:

$$\hat{S}_V = \left(|x_A - x_V| \leq k_C\right)\hat{S}_{V,C=1} + \left(|x_A - x_V| > k_C\right)\hat{S}_{V,C=2} \tag{11}$$

$$\hat{S}_A = \left(|x_A - x_V| \leq k_C\right)\hat{S}_{A,C=1} + \left(|x_A - x_V| > k_C\right)\hat{S}_{A,C=2} \tag{12}$$

where $|x_A - x_V|$ is the absolute audiovisual spatial disparity, and $k_C$ is the free parameter that represents the fixed criterion for integration (in degrees visual angle). Crucially, $k_C$ does not depend on stimulus uncertainty. If auditory information is highly uncertain (as in the case of spatial localisation), a large audiovisual disparity might still plausibly arise from a common source, but the Fixed Criterion model would nonetheless reject it (for more details, [55]). By contrast, if spatial disparity is explainable by noise, BCI may still infer a common cause.

The Stochastic Fusion (SF) model exhibits a lack of sensitivity to spatial disparity, similarly to the Forced Fusion model. However, Stochastic Fusion does not mandatorily perform integration. On each trial, it makes a random choice: either it fuses the audiovisual signals or it segregates them. Formally:

$$\hat{S}_V = (\eta > 0.5)\hat{S}_{V,C=1} + (\eta \leq 0.5)\hat{S}_{V,C=2} \tag{13}$$

$$\hat{S}_A = (\eta > 0.5)\hat{S}_{A,C=1} + (\eta \leq 0.5)\hat{S}_{A,C=2} \tag{14}$$

where $\eta$ is the free parameter that represents the probability of reporting a common cause. Crucially, $\eta$ does not depend on spatial disparity or sensory uncertainty. Nonetheless, Stochastic Fusion can partially accommodate scenarios where observers do not always integrate, because it sometimes performs segregation (for more details, [55]). Notably, while Stochastic Fusion incorporates a random decision process, Fixed Criterion leverages a priori information (via $k_C$) to guide the decision process, similar to Bayesian Causal Inference (via pcommon).

To determine which model architecture best explained each participant's localisation responses, and to evaluate the modulatory influence of action intention (communicative vs. non-communicative), we performed Bayesian model comparison in a 4 (BCI vs FC vs SF vs FF) × 2 ("Pooled" action intention conditions vs "Separated" conditions) factorial model space (S4a Fig; see also section "General Bayesian modelling approach" in the main text for details on model fitting and statistical analyses).

**Results.** In both experiments, Bayesian model comparison favoured the Fixed Criterion Separated model (S4b Fig; protected exceedance probability ≈ 1, Bayesian Omnibus Risk < 0.001). Focusing on the model parameters (S4c Fig), we first replicated the effects on the sensory noise parameters. We found no significant difference between the two action intention conditions for the visual noise $\sigma_V$ (Experiment 1: p = 0.204, es = 0.20, CI(95%) = [-0.02, 0.42], $BF_{10}$ = 0.77; Experiment 2: p = 0.505, es = -0.21, CI(95%) = [-0.58, 0.16], $BF_{10}$ = 0.35), while the auditory noise $\sigma_V$ was significantly higher in the non-communicative condition in Experiment 1 (p = 0.001, es = -0.83, CI(95%) = [-1.49, -0.16], $BF_{10}$ = 19.95) but not Experiment 2 (p = 0.104, es = 0.46, CI(95%) = [-0.39, 1.31], $BF_{10}$ = 0.55). Crucially, participants exhibited a more liberal criterion $k_C$ (i.e. integration response at higher spatial disparities) for the communicative than the non-communicative condition, specifically in Experiment 2 (Experiment 1: p = 0.865, es = 0.22, CI(95%) = [-0.90, 1.35], $BF_{10}$ = 0.19; Experiment 2: p = 0.003, es = 1.77, CI(95%) = [0.67, 2.88], $BF_{10}$ = 165.12; S4c Fig and S2 Table). Interestingly, in both experiments, model predictions were highly similar to those obtained with the BCI model (S2 and S3 Figs), and the two model architectures explained the same amount of variance ($R^2$ = 0.95 ± 0.01; S4 Table). Moreover, similar to the BCI results, individual increases of $w_{AV}$ for the communicative relative to the non-communicative condition ($w_{AV,Com} > w_{AV,NCom}$) positively correlated with the respective effect on the criterion ($k_{C,Com} > k_{C,NCom}$) and auditory noise ($\sigma_{A,Com} > \sigma_{A,NCom}$) in both experiments (Experiment 1: $k_C$: r = 0.69, p < 0.001; $\sigma_A$: r = 0.51, p = 0.002; Experiment 2: $k_C$: r = 0.78, p < 0.001; $\sigma_A$: r = 0.35, p = 0.044; S5 Fig).

The striking similarities between the BCI and Fixed Criterion results may suggest that they inherently capture the same underlying cognitive mechanism, that is, a stronger a priori belief that signals come from the same source under the communicative condition. If so, there should be a significant correlation between the BCI common cause prior pcommon and the Fixed Criterion $k_C$. Indeed, we found a highly significant correlation in both experiments (Experiment 1: r = 0.79, p < 0.001; Experiment 2: r = 0.76, p < 0.001; Fig 5). Collectively, these results indicate that observers solved audiovisual causal inference following non-optimal heuristics (i.e. changes in integration criterion) that nevertheless approximate optimal Bayesian inference with high accuracy. Crucially, a more liberal integration criterion for audiovisual communicative signals likely stems from a stronger prior expectation that these signals come from a common cause.

## Discussion

When we communicate face-to-face, we navigate a myriad of multisensory stimuli, often offset in time, that may or may not concur to convey shared meaning, raising a fundamental multisensory binding problem [4,5]. Yet, face-to-face communication allows for faster and more accurate information processing than speech alone, suggesting that bodily signals facilitate human communication [12–17]. Critically, despite clear evidence of multimodal facilitation, the underlying computational mechanisms remain largely unknown. The present study directly addressed this crucial, open question by testing competing computational accounts of multisensory integration during face-to-face communication. Using psychophysics, we quantified multisensory integration through the ventriloquist effect, which measures the extent to which a speaker's bodily signals influence the perceived location of their voice. Using Bayesian computational modelling, we then isolated the contribution of prior expectations and sensory uncertainty in shaping multisensory perceptual inference.

In a first analysis, we contrasted two normative accounts for how we parse the conversational scene into coherent multisensory units while segregating concurrent, yet unrelated, signals. On the one hand, observers may arbitrate between integration and segregation by taking into account the uncertainty about the underlying causal structure, as predicted by Bayesian Causal Inference [18,52–56]. In this framework, prior expectations would mediate whether sensory inputs

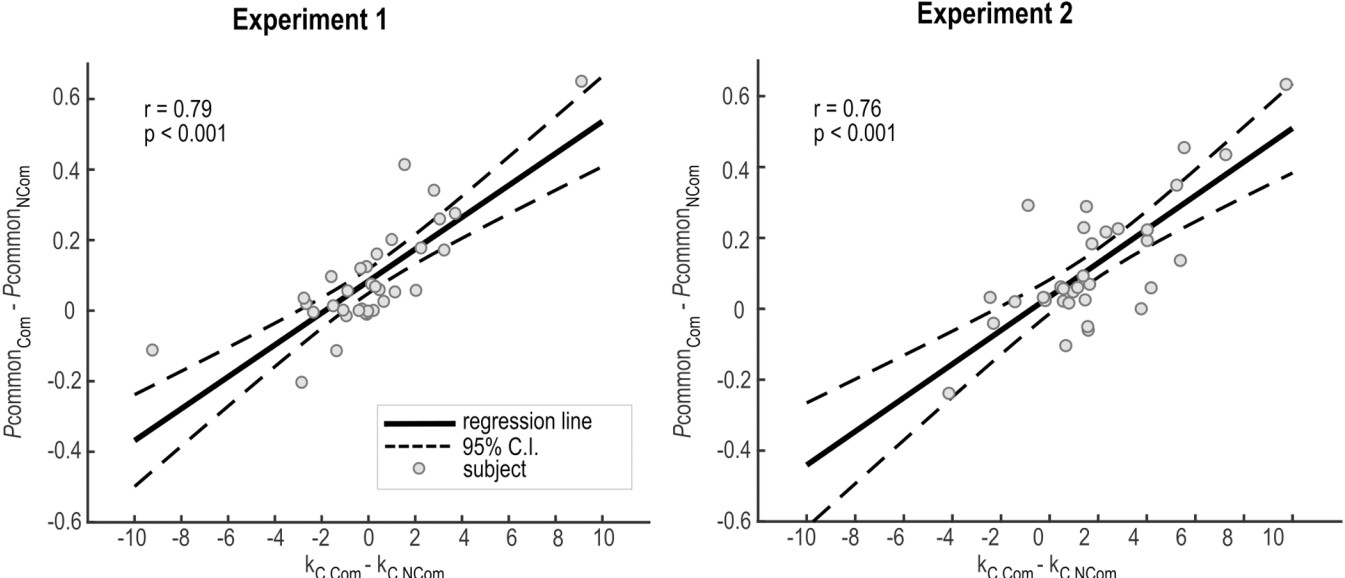

**Fig 5. Correlation between the BCI common-cause prior ($p_{common}$) and the criterion $k_C$.** Across participants' correlation between increases of $p_{common}$ for the communicative relative to the non-communicative condition ($p_{common,Com} − p_{common,NCom}$) and the respective increase of $k_C$. Pearson's correlation coefficients (r) and p-values (p) obtained from the two linear regression model: $p_{common} = β_0 + β_1 × k_C + ε$, where $β_0 =$ intercept, $β_1 =$ slope coefficient, $ε =$ residual error. Thick black line and dashed lines respectively show the linear regression line and the 95% confidence intervals (C.I.) from the linear regression models. Grey dots represent individual participants. Left and right plots show the results of Experiments 1 and 2 respectively.

are integrated into a unified percept or treated independently. On the other hand, observers may follow Forced Fusion principles [60–62], mandatorily integrating audiovisual signals regardless of causal uncertainty. Here, communicative bodily movements may attract greater attention than non-communicative ones due to higher perceptual salience [64–70] or social relevance [71,72]. Since attention decreases sensory uncertainty and thereby increases the weight of attended signals during the fusion process [73], communicative visual cues may guide the parsing of the conversational scene.

Across two consecutive experiments, our observations provide consistent and robust evidence in favour of Bayesian Causal Inference over Forced Fusion, demonstrating that prior expectations guide multisensory integration during face-to-face communication. In line with previous work [58,59,73,89], observers did not mandatorily integrate audiovisual signals, hence providing the same response irrespective of the task context; instead, the degree of integration was modulated by task relevance and spatial disparity. Crucially, observers showed a stronger a priori tendency to combine vocal and bodily signals when these indexed the same communicative intention (i.e., the speaker addresses the listener with their head, gaze and speech) compared to when this correspondence was absent. These findings demonstrate that pragmatic correspondences (i.e., alignment of communicative intention across modalities [1,4,6,7]) modulate the strength of the observer's common cause assumption, reinforcing the prior belief that audiovisual signals should be integrated into one unified percept [37–42]. While multiple cues typically concur to solve causal uncertainty, priority may be given to the most reliable cues depending on the characteristics of the perceptual scene and the current behavioural goal. During face-to-face communication, spatial and pragmatic correspondences may be a more reliable cue than temporal synchrony, given that vocal and bodily signals are often offset in time [4,5]. Notably, in the present study, we created pragmatic correspondences by introducing relevant linguistic content and ostensive bodily signals (head and gaze movements). Our design intentionally maximized the contrast between communicative and non-communicative signals to establish a clear proof-of-concept for the influence of communicative intent on audiovisual causal inference. While this approach introduces both low- and

high-level differences between conditions (such as stimulus dynamics, attentional demands, and semantic content), we view these as intrinsic features of real-world communicative contexts. In Experiment 2, we addressed one specific difference by equating stimulus durations across conditions, thereby reducing interpretive ambiguity. Future studies should incorporate more finely matched control conditions to better isolate the effects of communicative intent from other contributing factors such as attention, arousal, and stimulus complexity. For example, new control conditions could employ dynamic videos with gaze diverted from the addressee and utterances spoken in an unknown language. Furthermore, it would be interesting to extend the current findings to other relevant crossmodal cues, such as semantic correspondences [30–34] and socio-emotional cues [92]. Interestingly, we also found that prior expectations interacted with sensory uncertainty to determine the final perceptual estimates, in line with Bayesian Causal Inference principles [52,53]. In Experiment 1, multisensory integration was enhanced in the communicative condition due to a stronger common cause prior. Concurrently, the visual capture of sound was enhanced in the non-communicative condition due to higher auditory uncertainty. More specifically, higher auditory noise (stemming from the shorter duration of non-communicative stimuli) reduced the reliability of the auditory information, thereby increasing visual capture. When stimuli duration was equated across conditions in Experiment 2, removing differences in sensory uncertainty [90,91], we unveiled the direct influence of prior expectations on participants' behaviour.

In a follow-up analysis, we expanded the model space to include two non-optimal decision strategies offering psychologically plausible, heuristic approximations to perceptual inference [55]. First, the Fixed Criterion model applied a spatial disparity threshold to decide whether to integrate or segregate the audiovisual signals. Second, the Stochastic Fusion model randomly selected between integration and segregation on each trial. In both experiments, Fixed Criterion outperformed competing models in explaining participants' behaviour, although model predictions were highly similar to those produced by Bayesian Causal Inference. Crucially, participants exhibited a more liberal criterion (i.e., integration response at higher spatial disparities) when vocal and bodily inputs signalled the same communicative intention, as opposed to when this pragmatic correspondence was absent. Interestingly, this effect strongly correlated with an increase in the BCI common cause prior, suggesting that a more liberal integration criterion for audiovisual communicative signals may stem from a stronger prior expectation that these signals come from a common cause. Collectively, these results indicate that, at least in the presence of complex social stimuli, observers solve audiovisual causal inference following non-optimal heuristics (i.e., changes in integration criterion) that nevertheless approximate optimal Bayesian inference with high accuracy. Most importantly, the two computational strategies likely capture the same underlying cognitive mechanism: a stronger prior tendency to integrate vocal and bodily information when signals convey congruent communicative intent.

The present findings nicely dovetail with recent evidence that predictive processing may underlie multimodal facilitation in human communication [4,5,12,13,93]. Namely, we may process multimodal communicative signals more efficiently than speech alone because prior expectations guide our perception: we may parse the conversational scene into coherent meaningful units and in parallel predict upcoming congruent inputs as the interaction unfolds [5,94]. These observations raise fundamental questions on the origins of priors in multimodal communication. Given that face-to-face interactions represent the core ecological niche for language evolution, learning and use [1,2,95], one plausible hypothesis is that these priors are deeply rooted in the phylogenetic ethology [51] and developmental ecology [50] of human social interaction. In line with the "interaction-engine hypothesis" [96], humans may possess an evolutionary-shaped predisposition to infer communicative intent from multimodal signals, enabling efficient coordination in social exchanges. Such pragmatic priors may be partly innate and shared with our evolutionary ancestors [1,97], providing a cognitive foundation for early sensitivity to multimodal communicative cues, such as gaze direction, pointing and infant-directed speech [98–101]. Notably, these cues catalyze and scaffold the transmission of cultural knowledge from the first year of life onward in humans [102,103]. However, the precise mechanisms by which these priors emerge, adapt and specialise across development require further investigation. For instance, are these priors universally present across cultures or do they vary with differences in communicative conventions (e.g., eye contact conventions in Western versus Eastern cultures [104,105])? If

multimodal priors are also tuned by experience, what are the critical periods for their plasticity? Answering these questions will require cross-cultural, developmental and comparative research approaches. These endeavours will not only advance our understanding of human communication but also inform artificial intelligence and robotics, helping to design systems that interpret communicative intent in a biologically plausible fashion.

In conclusion, the present study combines psychophysics with Bayesian modelling to provide direct and compelling evidence that prior expectations guide multisensory integration during face-to-face interactions. These findings indicate that multimodal facilitation in human communication does not solely rely on the redundancy of congruent multisensory signals but is also critically shaped by deeply ingrained expectations about how communicative signals should be structured and interpreted.

## Supporting information

**S1 Text. Response times analysis and results.**
(DOCX)

**S1 Fig. Response times.** Across-participants' mean (± SEM) response times in Experiments 1 and 2, as a function of response modality (auditory; visual), action intention (communicative: Com; non-communicative: NCom) and spatial disparity (0°: NoDisp; 9°: LowDisp; 18°: HighDisp).
(TIF)

**S2 Fig. Distributions of spatial estimates.** Distributions of spatial estimates in Experiments 1 and 2 given by participants' localization responses (solid lines) or predicted by the BCI Separated model (dashed line) and FC Separated model (dotted line) fitted to each participant's responses as a function of response modality (auditory: A; visual: V), action intention (communicative: Com; non-communicative: NCom) and stimuli position (0, 9, 18 degrees visual angle). In the 3 × 3 subplots, the visual position is represented on the y-axis and the auditory position is represented on the x-axis. For each audiovisual combination (each subplot), the 3 possible answers are represented on the x-axis (left, centre, right) and participants' proportion of responses for each of the possible answers is represented on the y-axis.
(TIF)

**S3 Fig. Audiovisual weight index ($w_{AV}$) sensitivity analysis with empirical and predicted data. a)** Across participants' mean ± SEM empirical $w_{AV}$ obtained from the Experiments, **b)** predicted $w_{AV}$ obtained from model predictions of the BCI Separated model and **c)** predicted $w_{AV}$ obtained from model predictions of the FC Separated model. Results are plotted as a function of action intention (communicative: Com; non-communicative: NCom), response modality (auditory: A; visual: V) and audiovisual spatial disparity (9°: LowDisp; 18°: HighDisp). An index equal to 1 (respectively, 0) indicates pure visual (respectively, auditory) influence on participants' localization responses; values between 0 and 1 indicate intermediate degrees of audiovisual integration. General bias: $w_{AV}$ obtained using the reported location in audiovisual congruent trials averaged across participants and experimental conditions (action intention × response modality). Specific bias: $w_{AV}$ obtained using the participant-specific and condition-specific average reported location in audiovisual congruent trials.
(TIF)

**S4 Fig. Factorial space and modelling results to assess non-optimal decision strategies. a)** To determine whether non-optimal decision strategies best explained each participant's localisation responses, and to evaluate the modulatory influence of action intention (communicative vs. non-communicative), we performed Bayesian model comparison in a 4 (BCI vs FC vs SF vs FF) × 2 (Pooled vs Separated action intention conditions) factorial model space. **b)** Protected exceedance probability (pEP, grayscale) of each model in the 4 × 2 model comparison factorial space, i.e., how likely each model is to explain the data compared to the other models. Bayesian Omnibus Risk (BOR) represents the probability that the results are due to chance. **c)** Across participants' mean ± SEM parameter estimates of fixed criterion ($k_C$), auditory noise ($\sigma_A$), and

visual noise ($\sigma_V$) of the winning model (i.e., FC Separated) as a function of action intention. p-values based on two-tailed between-subject permutation tests. Top and bottom rows show the results of Experiment 1 and 2 respectively.
(TIF)

**S5 Fig. Correlation between audiovisual weight index ($w_{AV}$) and Fixed Criterion model parameters.** Across participants' correlation between increases of $w_{AV}$ for the communicative relative to the non-communicative condition ($w_{AV,Com} - w_{AV,NCom}$) and the respective effect on a) the common cause prior ($k_{C,Com} - k_{C,NCom}$) and b) the auditory noise ($\sigma_{A,Com} - \sigma_{A,NCom}$) of the winning model (FC Separated). Pearson's correlation coefficients (r) and p-values (p) obtained from the two linear regression models: (1) $w_{AV} = \beta_0 + \beta_1 \times k_C + \varepsilon$ and (2) $w_{AV} = \beta_0 + \beta_1 \times \sigma_A + \varepsilon$, where $\beta_0$ = intercept, $\beta_1$ = slope coefficient, $\varepsilon$ = residual error. Thick black line and dashed lines respectively show the linear regression line and the 95% confidence intervals (C.I.) from the linear regression models. Grey dots represent individual participants. Top and bottom rows show the results of Experiments 1 and 2 respectively.
(TIF)

**S1 Table. Audiovisual weight index ($w_{AV}$): descriptive statistics.** Across-participants' mean (± SEM) *wAV* as a function of action intention (communicative: Com; non-communicative: NCom), response modality (repA: auditory; repV: visual) and audiovisual spatial disparity (9°: LowDisp; 18°: HighDisp) for Experiments 1 and 2.
(DOCX)

**S2 Table. Bayesian models parameters.** Across participants' mean (±SEM) of the models' parameters: $p_{common}$, common-cause prior probability; $k_C$, fixed criterion (° visual angle); η, probability of fusion response; $\sigma_P$, spatial prior standard deviation (° visual angle); $\sigma_A$, auditory likelihood standard deviation (° visual angle); $\sigma_V$, visual likelihood standard deviation (° visual angle). Model architectures: Bayesian Causal Inference (BCI); Fixed Criterion (FC); Stochastic Fusion (SF); Forced Fusion (FF). While "Pooled" models do not account for the influence of action intention (communicative: Com vs. non-communicative: NCom), "Separated" models have separate parameters for each action intention condition.
(DOCX)

**S3 Table. Bayesian model comparison (4 models).** Model architectures: Bayesian Causal Inference (BCI); Forced Fusion (FF). To determine which model architecture best explained each participant's localisation responses, and to evaluate the modulatory influence of action intention (communicative vs. non-communicative), we performed random-effect Bayesian model comparison in a 2 (BCI vs FF) × 2 ("Pooled" action intention conditions vs "Separated" conditions) factorial model space. We report across participants' mean (±SEM) coefficient of determination ($R^2$) and protected exceedance probability (*pEP*; probability that a model is more likely than the other models, beyond differences due to chance).
(DOCX)

**S4 Table. Bayesian model comparison (8 models).** Model architectures: Bayesian Causal Inference (BCI); Fixed Criterion (FC); Stochastic Fusion (SF); Forced Fusion (FF). To determine which model architecture best explained each participant's localisation responses, and to evaluate the modulatory influence of action intention (communicative vs. non-communicative), we performed random-effect Bayesian model comparison in a 4 (BCI vs FC vs SF vs FF) × 2 ("Pooled" action intention conditions vs "Separated" conditions) factorial model space. We report across participants' mean (±SEM) coefficient of determination ($R^2$) and protected exceedance probability (*pEP*; probability that a model is more likely than the other models, beyond differences due to chance).
(DOCX)

**S5 Table. Response times: descriptive statistics.** Across participants' mean (±SEM) response times as a function of action intention (communicative: Com; non-communicative: NCom), response modality (repA: auditory; repV: visual) and audiovisual spatial disparity (0°: NoDisp; 9°: LowDisp; 18°: HighDisp) for Experiments 1 and 2.
(DOCX)

**S6 Table. Response times: ANOVA results.** Main effects and interactions for the response times in the 2 (action intention: communicative; non-communicative) × 2 (response modality: auditory; visual) × 3 (spatial disparity: none; low; high) repeated measures ANOVA. Greenhouse-Geisser correction is applied in case of violation of sphericity (Mauchly's test). (DOCX)

**S7 Table. Response times: post-hoc comparisons.** Post-hoc comparisons for the significant two-way interactions of the response times ANOVA: Action Intention × Response Modality; Action Intention × Spatial Disparity; Response Modality × Spatial Disparity. Com: communicative action; NCom: non-communicative action; Aud: auditory response; Vis: visual response; NoDisp: no spatial disparity (0°); LowDisp: low disparity (9°); HighDisp: high disparity (18°). P-values were adjusted using the Holm correction. (DOCX)

**S8 Table. Audiovisual weight index ($w_{AV}$) sensitivity analysis.** Across participants' mean (±SEM) $w_{AV}$ as a function of action intention (Comm: communicative; NonCom: non-communicative), response mdoality (repA: auditory; repV: visual) and audiovisual spatial disparity (LowDisp: 9° visual angle; HighDisp: 18° visual angle) for Experiments 1 a) and 2 b). The $w_{AV}$ was computed using the average reported location for congruent trials specific for each participant in the four experimental conditions (Action × Response modality). (DOCX)

**S9 Table. Audiovisual weight index ($w_{AV}$) sensitivity analysis: statistical significance.** Main effects and interactions for the audiovisual weight index ($w_{AV}$) in Experiments 1 and 2 in the 2 (action intention: communicative vs. non-communicative) × 2 (response modality: auditory vs. visual) × 2 (audiovisual disparity: low vs. high) factorial design. The $w_{AV}$ was computed using the average reported location for congruent trials specific for each participant in the four experimental conditions (Action × Response modality). P-values are based on two tailed permutation tests. Effect sizes [95% CI] correspond to the difference of the across participants' mean empirical effect and the mean of the non-parametric null-distribution. (DOCX)

## Acknowledgments

The authors thank David Meijer for helpful discussion on the computational modelling approach.

## Author contributions

**Conceptualization:** Giulia Mazzi, Ambra Ferrari, Stefania Benetti.

**Data curation:** Giulia Mazzi, Ambra Ferrari.

**Formal analysis:** Giulia Mazzi, Ambra Ferrari.

**Funding acquisition:** Francesco Pavani, Stefania Benetti.

**Investigation:** Giulia Mazzi, Maria Laura Mencaroni, Chiara Valzolgher, Mirko Tommasini.

**Methodology:** Giulia Mazzi, Ambra Ferrari.

**Project administration:** Giulia Mazzi, Ambra Ferrari, Stefania Benetti.

**Software:** Giulia Mazzi, Ambra Ferrari.

**Supervision:** Stefania Benetti.

**Validation:** Giulia Mazzi, Ambra Ferrari.

**Visualization:** Giulia Mazzi, Ambra Ferrari.

**Writing – original draft:** Giulia Mazzi, Ambra Ferrari.

**Writing – review & editing:** Giulia Mazzi, Ambra Ferrari, Maria Laura Mencaroni, Chiara Valzolgher, Mirko Tommasini, Francesco Pavani, Stefania Benetti.

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
