## [Decision Letter · Decision Letter 0]

21 Apr 2025

Prior expectations guide multisensory integration during face-to-face communication

PLOS Computational Biology

Dear Dr. Mazzi,

Thank you for submitting your manuscript to PLOS Computational Biology. After careful consideration, we feel that it has merit but does not fully meet PLOS Computational Biology's publication criteria as it currently stands. Therefore, we invite you to submit a revised version of the manuscript that addresses the points raised during the review process.

Please submit your revised manuscript within 60 days Jun 21 2025 11:59PM. If you will need more time than this to complete your revisions, please reply to this message or contact the journal office at ploscompbiol@plos.org. Please include the following items when submitting your revised manuscript:

We look forward to receiving your revised manuscript.

Kind regards,

Zhiyi Chen

Academic Editor

PLOS Computational Biology

Lyle Graham

Section Editor

PLOS Computational Biology

**Additional Editor Comments :**

Thank you for the patience. We have received comments from three peer experts. Despite a recommendation to reject this manuscript, I perceived that this comment is not adequately constructive and clear. As Reviewer #1 and #3 recommended, along with my personal evaluation, I recommend to invite a major revision to address all the structural and clear concerns shared by them.

**Journal Requirements:**

- ® on page: 8.

5) Figure 1 includes an image of an identifiable person. Please provide written confirmation or release forms, signed by the subject(s) (or their guardian), giving permission to be photographed and to have their images published under a Creative Commons license. You may upload permission forms to your submission file inventory as item type 'Other'. Otherwise, we kindly request that you remove the photograph.

Potential Copyright Issues:

i) Please confirm (a) that you are the photographer of 1A, and 1C, or (b) provide written permission from the photographer to publish the photo(s) under our CC BY 4.0 license.

ii) Figures 1A, 1C, and 2A. Please confirm whether you drew the images / clip-art within the figure panels by hand. If you did not draw the images, please provide (a) a link to the source of the images or icons and their license / terms of use; or (b) written permission from the copyright holder to publish the images or icons under our CC BY 4.0 license. Alternatively, you may replace the images with open source alternatives. See these open source resources you may use to replace images / clip-art:

7) Your current Financial Disclosure states receiving many funds. However, your funding information on the submission form indicates only one fund. Please ensure that the funders and grant numbers match between the Financial Disclosure field and the Funding Information tab in your submission form. Note that the funders must be provided in the same order in both places as well.

**Reviewers' comments:**

Reviewer's Responses to Questions

Reviewer #1: Thank you for the opportunity to review this manuscript. I really enjoyed reading it!

The study presents a well-structured and compelling investigation into multimodal integration in face-to-face communication, combining computational and experimental approaches. Specifically, the authors test whether Hierarchical Bayesian Causal Inference (HBCI) or Forced Fusion better explains human multimodal processing. I appreciate the authors' effort in using sophisticated methods to address such an important and complex question about the underlying mechanisms of multimodal communication. The topic is also both interesting and timely. However, I have some concerns related to the experimental design and stimuli, which I describe below.

1. The duration of communicative and non-communicative signals differs between experiments (1500ms vs. 500ms). The authors acknowledge that this likely led to higher auditory noise in the non-communicative condition in Exp. 1. While this was addressed in Exp. 2 by adjusting the duration of the non-communicative stimuli, it raises the question of why it was not controlled from the beginning. Although the authors mention their intention to use these materials in future neuroimaging studies, the duration difference remains a potential confound. Would it make sense to focus on Exp. 2 in the main text and summarize Exp. 1 in the SM instead? This is just a suggestion, and the authors may decide against it if they believe it does not improve the manuscript.

2. The study would benefit from additional discussion on the potential impact of stimulus properties, such as duration and the positioning of auditory and visual cues. Do we know whether—and if so, how—these angles influence processing and integration? While conducting new norming experiments may not be feasible at this stage, grounding these choices more explicitly in prior literature could strengthen the rationale. Similarly, the study assumes that shifting recorded videos (e.g., from right to left) effectively simulates (in)direct eye contact. However, subtle inconsistencies in gaze and speaker orientation might introduce unintended cues that affect multisensory integration. I am also not convinced that these manipulations are truly more naturalistic (e.g., how often does the speaker entirely avoid eye contact with their interlocutor in a natural conversation? When it does occur, it is often due to conversational interruptions or because the message is (re)directed at someone else). Providing further justification for this approach could help clarify its validity.

3. I like the idea of including a word association task at the end of communicative blocks. However, it introduces another potential confound that could influence the results. For example, it is possible that semantic or pragmatic expectations are strengthening the common cause prior, rather than audiovisual integration alone. Would it have been beneficial (if possible at all) to include a comparable semantic task in the non-communicative condition to allow for a more direct comparison?

4. I was wondering why the authors decided to use still frames and vowel vocalization in the non-communicative condition, rather than videos of non-communicative gestures with naturalistic vocalizations (e.g., yawning or coughing). Using such more naturalistic non-communicative stimuli might have led to a more balanced comparison.

5. From the figures, it appears that when an auditory stimulus was heard from the right, the headphones were displayed on the right, and similarly, when the speaker looked from the right, her video was also positioned on the right (and similarly for the left side and middle locations). Could the authors clarify the rationale for this design choice? An alternative approach might have been to present a central video with audio playing in the background. If the stimuli were always positioned congruently, participants might not need to truly integrate the signals but rather just recall their spatial locations. If I have misunderstood the trial structure, I apologize—please consider clarifying this in the figure or text to ensure other readers are not similarly confused.

I also have some additional discussion points that the authors may wish to consider.

6. The authors argue that enhanced attention due to communicative salience could still be accounted for by Forced Fusion, as increased reliability strengthens integration. While the study suggests that HBCI uniquely explains the findings, was an alternative Forced Fusion model with an attention-weighted reliability component explicitly ruled out? Addressing this could further solidify the conclusions.

7. A more conceptual question—if non-communicative signals (or noise) are not part of the conversation (i.e., they are not integrated) in the first place, is there a strong justification for modeling them within a multisensory integration framework?

Finally, I have a minor suggestion, if the journal guidelines allow and the authors agree: a table summarizing key terminology (e.g., ventriloquist effect, HBCI, Forced Fusion, etc.) might be a helpful addition for readers less familiar with the field.

Reviewer #2: The present study is interesting, but the structure of the manuscript is not consistent with PLOS Computational Biology. For example, the results are presented in two sections, or the conclusion section is missing. Also, the discussion section needs substantial strengthening.

Reviewer #3: Mazzi Ferrari 2025 Plos Comp Biol

Prior expectations guide multisensory integration during face-to-face communication

In two psychophysical experiments, the study investigates whether near-natural audiovisual communication signals are integrated or segregating depending on their communicative intent. Using computational modelling, the authors report that integration/segregation follows the principles of Bayesian causal inference (BCI), which has been so far mostly applied to simplified non-social audiovisual signals. More specifically, the author find that commucitive signals (compared to a non-communicative control conditions) increase the BCI model’s causal prior parameter.

Overall, the study investigates a very interesting and urgent research question (causal inference on social AV signals), applies well implemented experimental stimuli and design, and thoroughly models and analyses behavioral data. The manuscript is very well written and nice to read. A further strength is that the authors performed two experiments, the second to address a shortcoming of the first experiment (which was unequal duration of A stimuli between communication conditions) and to replicate results from the first experiment.

My major concern is the implementation of the control condition: While the experimental condition with communicative signals uses a video with meaningful speech signals and semantic secondary task, the non-communicative control condition only used static pictures with vowel sounds. This leaves both conditions unmatched in many aspects beyond ‘communicative’ signals (stimulus dynamics, differential secondary task etc.). This introduces many potential confounds (e.g. attention/salience) which are known to also influence audiovisual integration and specifically the causal prior.

My more specific comments:

Methods:

• Auditory spatial stimuli: The authors used HRTF to create virtual spatial sounds. Which head model did they use? How well were the A spatial stimuli localizable?

• Control condition of static picture with a single auditory vowel and without secondary task: This choice introduces many low level (e.g. visual motion) and higher-level (e.g. attention/saliency, dual task demands, arousal etc.) differences compared to the dynamic video in the experimental condition. Further, the non-communicative pictures are much shorter than videos (500 ms vs. 1-2 s), which was corrected in Exp. 2. Because communicate videos even conveyed a short “story” over trials in a block, there is even a semantic difference. While this is a creative idea to increase participants’ motivation, it also makes the control condition even less comparable. Overall, this choice introduces many low- and high-level confounds, which are partially already known to influence AV integration and also BCI model parameters. For example, attention affects the causal prior (Badde 2020) and variance parameters (their own study, Ferrari & Noppeney 2021), so that differential attentional processes may also explain the results. So I think a more comparable control conditions would have allowed to more specific conclusions whether really the social signals make the difference. This may be even more important for a future neuroimgaging study (which seems to be planned).

• Audiovisual weight index:

o Why was the wAV normalized by the average reported location in audiovisual congruent trials computed across participants? The normalization helps to account for individual response biases (e.g. a central tendency), but when computing it with the across-participants average, individual response biases are not accounted for. I would suggest to make a sensitivity analyses whether this choice (individual vs. group-mean response location) affects the results.

o wAV is analysed with non-parametric permutation tests, but important details how this permutation test was implemented is missing (e.g. permutation scheme for main/interaction effects, test statistic used etc.)

• BCI modelling: The authors only compare the BCI model to a forced fusion model, which is a rather weak model when spatial disparities are introduced (because the model’s common cause assumption is obviously wrong). Stronger / equally complex models are stochastic fusion and fixed criterion model (Acerbi et al. 2018 Plos Comp Biol), the latter can also accommodate spatial disparity in a (non-optimal) fashion. Thus, the authors could expand their model space. An additional interesting idea to explore with more model candidates would be that participants may also switch their perceptual decision strategy (e.g. from causal inference to stochastic fusion) depending on the communicative signals.

• The non-parametric comparison of model parameters could be implemented with a permutation test, which should provide better power compared to a Wilcoxon signed-rank test.

• If possible, the authors may consider reporting Bayes factors where applicable to provided also evidence for null effects (e.g. the parameter comparison)

• I highly appreciate that the authors share documented code and data (good open science practices!).

Results:

• Exp. 1: I find it surprising that the communicative condition does not modulate wAV, but it strongly affects the causal prior and the auditory variance model parameters. Why do we not see this in the wAV index? E.g. a higher causal prior in the communicative condition should translate into a higher wAV for A report. This inconsistency deserves further exploration…I think the author should also plot model predictions for wAV to understand this inconsistency. It is even more striking because higher causal prior (= more V bias) and lower auditory variance (= more V bias) for communicative signals influence wAV in the same direction for A report, towards stronger V influence (as found in Exp. 2). So, I agree with the author’s BCI model interpretation that a larger causal prior increases multisensory integration for both A and V report, but I disagree with the authors’ interpretation that larger auditory variance increases multisensory integration in general: larger auditory variance generally increases the visual / decreases the auditory influence for both A and V report (i.e. a vertical shift in wAV for both tasks). For A report, this increased wAV can be interpreted as stronger integration, but for V report, the (anyhow tiny) A influence is even further reduced, which means stronger segreation (theoretically within the BCI model). Thus, shifts in A or V variance do not generally increase / decrease multisensory integration.

• Exp. 2: It is good to see that now wAV and parameter comparison align, when the duration of the A stimulus in both communication conditions is matched. One way to test this ‘parameter – wAV ’ link more formally would be to correlate with individual causal priors / auditory variances with the individual communication signal effect on wAV (also in Exp. 1). As above, I would not state that higher auditory noise enhances multisensory integration in general (only for A report this makes sense because the stronger V bias is a signature of MI)

Discussion:

• I think the authors need to acknowledge the limitation that the communicative videos could increase the common cause prior due to confounds/unspecific mechanisms, such as attention/saliency/arousal/dual task demands etc.. The authors state that they wanted to maximize the difference between the communicative and non-communicative conditions, but I think this decision comes at the price of many potential confounds, quite strongly limiting the specificity of their conclusions.

Minor:

• Intro, l. 86: “guided by the amount of audiovisual spatial disparity and thus the strength of the common cause prior”. The “thus” sounds a bit misleading, spatial disparity and common cause prior are independent (because the common cause prior is a prior and not the posterior causal probability)

• wAV is (I feel) more often called crossmodal bias, and has been introduced (to my knowledge) by Wallace et al. 2004

• L. 405 and Table 1: A probability from a statistical test is never p = 0, but e.g. p < 0.001

**Have the authors made all data and (if applicable) computational code underlying the findings in their manuscript fully available?**

Reviewer #1: Yes

Reviewer #2: None

Reviewer #3: Yes

PLOS authors have the option to publish the peer review history of their article (what does this mean? ). If published, this will include your full peer review and any attached files.

**Do you want your identity to be public for this peer review?** For information about this choice, including consent withdrawal, please see our Privacy Policy .

Reviewer #1: No

Reviewer #2: No

Reviewer #3: **Yes: ** Tim Rohe

**Figure resubmission:**
---

## [Decision Letter · Decision Letter 1]

27 Jul 2025

PCOMPBIOL-D-25-00334R1

Prior expectations guide multisensory integration during face-to-face communication

PLOS Computational Biology

Dear Dr. Mazzi,

Thank you for submitting your manuscript to PLOS Computational Biology. After careful consideration, we feel that it has merit but does not fully meet PLOS Computational Biology's publication criteria as it currently stands. Therefore, we invite you to submit a revised version of the manuscript that addresses the points raised during the review process.

Please submit your revised manuscript within 30 days Sep 26 2025 11:59PM. If you will need more time than this to complete your revisions, please reply to this message or contact the journal office at ploscompbiol@plos.org. Please include the following items when submitting your revised manuscript:

We look forward to receiving your revised manuscript.

Kind regards,

Zhiyi Chen

Academic Editor

PLOS Computational Biology

Lyle Graham

Section Editor

PLOS Computational Biology

**Additional Editor Comments:**

Thank you for addressing reviewer's concerns well. As the Reviewer #3 remained, I would recommend authors to fully address these minor concerns.

**Reviewers' comments:**

Reviewer's Responses to Questions

Reviewer #1: I would like to thank the authors for their clear and thorough responses. They addressed both my concerns regarding the clarity of the text and the more conceptual issues I raised. I believe they have done an excellent job in refining and polishing the manuscript. This is a very interesting and valuable piece of work, and I look forward to seeing it in print!

Reviewer #3: Overall, the authors adress my concerns very well, and their clarifications and additional analyses strengthen the manuscript. My major remaining questions pertains to the comparison of the BCI model to the stochastic fusion and fixed criterion models:

• The authors show that their auditory spatial stimuli work as intended, using generic HRTFs from a database. Still, I don’t understand which of the HRTFs was ‘relevant’ and used in the study (e.g. averaged HRTF, matched HRTFs)? Absolute detail, but important for replication purposes.

• Confounds in control condition: The authors now acklowledge and discuss that the visual static control condition differs in many low- and high-level aspects from the experimental condition. I agree with the authors that ‘communicate intent’ is probably a high-level concept that varies on more than a single sensory dimension or quality when implementing experimentally with some ecological validity. So, it is exactly the usual dilemma between internal and external/ecological valdity. Still, I think better control conditions are conceivable (e.g. dynamic videos with unknown language and gaze diverted from the participant). However, the authors should acknowledge this choice (or limitation) throughout the manuscript, e.g. when motivation the stimuli (demonstrating that it is feature and not a bug).

• Results account now for indivdiual response biases.

• The authors justify in length why the stochastic fusion model and fixed criterion models are not suitably for their research question. But this is not very convincing to me, both are plausible and stronger model competitors to BCI. And I think specifically in the context of more complex social signals (compare to the usual flashes and beeps), I would be interesting to learn whether observers adopt non-optimal approximate heuristics. So in this case I think this does not need to be deferred to future studies, but can be implemented in the current study. Even if these models turn to be stronger than BCI, this would not make the study less interesting and important.

• The authors updated their statistics with permutation tests and Bayes factors as suggested, improving statistical rigor.

• The authors now explain the counteracting effect of causal prior and auditory variance on wAV (which might be very counterintuiative to non-expert readers) and show positive correlations between model parameters and experimental effects for both experiments. This strengthens their findings and link of computational model and behavioral data!

**Have the authors made all data and (if applicable) computational code underlying the findings in their manuscript fully available?**

Reviewer #1: None

Reviewer #2: None

Reviewer #3: Yes

PLOS authors have the option to publish the peer review history of their article (what does this mean? ). If published, this will include your full peer review and any attached files.

**Do you want your identity to be public for this peer review?** For information about this choice, including consent withdrawal, please see our Privacy Policy .

Reviewer #1: No

Reviewer #2: No

Reviewer #3: No

**Figure resubmission:**
---

## [Editor Report · Decision Letter 2]

27 Aug 2025

Dear Ms. Mazzi,

We are pleased to inform you that your manuscript 'Prior expectations guide multisensory integration during face-to-face communication' has been provisionally accepted for publication in PLOS Computational Biology.

Best regards,

Zhiyi Chen

Academic Editor

PLOS Computational Biology

Lyle Graham

Section Editor

PLOS Computational Biology

Authors addressed all the concerns that all the reviewers raised. I recommend to accept this manuscript in the current form.

---

## [Editor Report · Acceptance letter]

PCOMPBIOL-D-25-00334R2

Prior expectations guide multisensory integration during face-to-face communication

Dear Dr Mazzi,

I am pleased to inform you that your manuscript has been formally accepted for publication in PLOS Computational Biology. Your manuscript is now with our production department and you will be notified of the publication date in due course.

With kind regards,

Zsuzsanna Gémesi
